# Beyond Chemical QA: Evaluating LLM's Chemical Reasoning with Modular Chemical Operations

**Hao Li**[1,2,3,4,◇,★]   **He Cao**[2,◇]   **Bin Feng**[2]   **Yanjun Shao**[5]   **Xiangru Tang**[5]   **Zhiyuan Yan**[3,4,†]
**Yonghong Tian**[1,3,4,†]   **Li Yuan**[1,3,4,†]   **Yu Li**[2,†,‡]

[1]Pengcheng Laboratory   [2]International Digital Economy Academy
[3]School of Electronic and Computer Engineering, Peking University
[4]School of AI for Science, Peking University   [5]Yale University

https://huggingface.co/datasets/OpenMol/ChemCoTBench
https://huggingface.co/datasets/OpenMol/ChemCoTBench-CoT
https://github.com/IDEA-XL/ChemCoTBench/

## Abstract

While large language models (LLMs) with Chain-of-Thought (CoT) reasoning excel in mathematics and coding, their potential for systematic reasoning in chemistry, a domain demanding rigorous structural analysis for real-world tasks like drug design and reaction engineering, remains untapped. Current benchmarks focus on simple knowledge retrieval, neglecting step-by-step reasoning required for complex tasks such as molecular optimization and reaction prediction. To address this, we introduce ChemCoTBench, a reasoning framework that bridges molecular structure understanding with arithmetic-inspired operations, including addition, deletion, and substitution, to formalize chemical problem-solving into transparent, step-by-step workflows. By treating molecular transformations as modular "chemical operations", the framework enables slow-thinking reasoning, mirroring the logic of mathematical proofs while grounding solutions in real-world chemical constraints. We evaluate models on two high-impact tasks: Molecular Property Optimization and Chemical Reaction Prediction. These tasks mirror real-world challenges while providing structured evaluability. We further provide ChemCoTDataset, a pioneering 22,000-instance chemical reasoning dataset with expert-annotated chains of thought to facilitate LLM fine-tuning. By providing annotated trainable datasets, a reasoning taxonomy, and baseline evaluations, our work bridges the gap between abstract reasoning methods and practical chemical discovery, establishing a foundation for advancing LLMs as tools for AI-driven scientific innovation.

## 1 Introduction

With the rapid advancement of large language models (LLMs), reasoning capabilities have become a defining measure of performance. Techniques like chain-of-thought [67] prompting enable LLMs to decompose complex problems into structured, human-like reasoning steps (**system-II** [30]), achieving breakthroughs in mathematics [50, 57, 71], coding [14, 23], and even Olympiad-level challenges [17, 22, 64]. Despite recent advances in LLM reasoning capabilities, chemistry, a discipline fundamental to areas like drug discovery and materials science, still lacks a benchmark that assesses whether these improvements extend to its complex, domain-specific problem-solving needs. While several benchmarks have been proposed for LLMs in chemistry [16, 35, 40, 45, 73], they primarily focus on domain-specific question answering, which suffers from several key limitations:

---

◇ Equal contributors, ★ Work done as a student researcher at IDEA.
† Corresponding Authors, ‡ Team Leader, Connect Email: lihao1984@pku.edu.cn

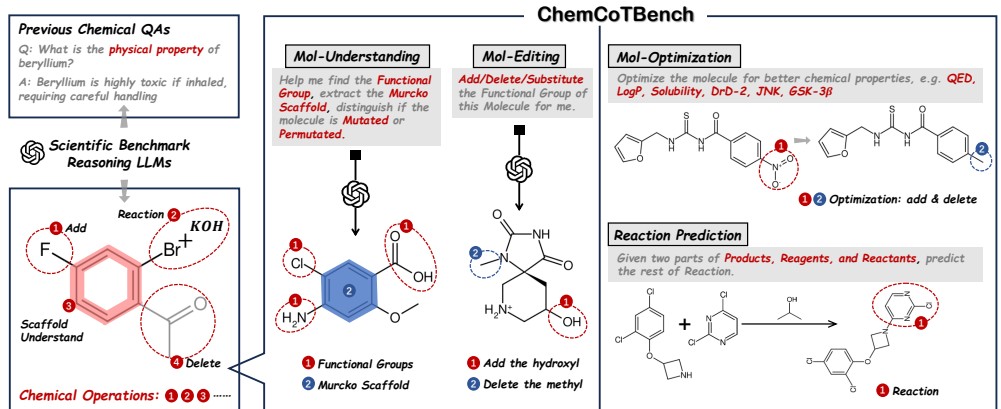

Figure 1: Previous chemical benchmarks focus on factual recall with domain knowledge, while our ChemCoTBench focuses on the evaluation of step-wise reasoning for complex chemical problems by defining a set of modular chemical operations.

**1. Lack of Structured, Stepwise Reasoning and Real-World Relevance:** Current evaluations often reduce chemistry assessment to factual recall (e.g., naming compounds or reactions), neglecting the need for operational reasoning akin to arithmetic or coding. Unlike mathematical problems, where solutions demand explicit, verifiable steps, chemistry QA tasks fail to simulate how experts decompose challenges. For instance, they don't capture the process of iteratively refining a molecule's substructure to optimize properties, considering crucial real-world factors like synthesizability or toxicity, or deducing reaction mechanisms through intermediate transformations. This gap means we're not fully evaluating the analytical depth required in real-world chemistry. Therefore, evaluations must shift from these textbook-like problems to challenges that better reflect practical applications.

**2. Ambiguous Skill Attribution in Hybrid Evaluations:** Existing benchmarks [39, 53, 66] often conflate reasoning, knowledge recall, and numerical computation into single "exam-style" metrics—for instance, asking LLMs to calculate reaction yields while simultaneously recalling reagent properties. This obscures whether strong performance stems from structured reasoning (e.g., analyzing reaction pathways) or memorized facts (e.g., solvent boiling points). Such ambiguity hinders targeted model improvement and misaligns evaluations with downstream tasks like drug discovery, where success depends on modular reasoning (e.g., decoupling molecular design from synthesizability checks) rather than monolithic problem-solving.

To address these limitations, we introduce **ChemCoTBench**, a **step-by-step**, **application-oriented**, and **high-quality** benchmark for evaluating LLM reasoning in chemical applications. A core innovation of ChemCoTBench is its formulation of complex chemical tasks, specifically targeting molecular modeling and design (Fig.1), into explicit sequences of verifiable modular chemical operations on SMILES structures (e.g., substructure addition, deletion, or substitution). This approach allows for a granular assessment of an LLM's ability to execute and chain together fundamental chemical transformations. The benchmark features progressively challenging tasks, spanning from basic molecular understanding and editing to property-guided structure optimization and complex multi-molecule chemical reactions. **High-quality** evaluation is ensured through a dual validation process combining LLM judgment with expert review from 13 chemists. Furthermore, **ChemCoTDataset** is introduced as the first chemical reasoning dataset with precise chain-of-thought labels. Its 22,000 instances facilitate effective fine-tuning of Large Language Models.

We evaluate the chemical reasoning ability across reasoning-enhanced and non-reasoning LLMs. Experimental results reveal room for improvement in reasoning LLMs, particularly open-source and distilled-reasoning LLMs, when addressing complex chemical problems. While these models demonstrate strong performance in complex mathematical and coding tasks, they are unable to organize chemical knowledge and establish step-wise modular chemical operations due to the scarcity of chemical reasoning data. Notably, ChemCoTDataset, the large chemical CoT dataset provided by ChemCoTBench, is shown to enhance chemical reasoning performance, effectively addressing the reasoning data scarcity issue in chemical reasoning domain for LLMs.

To summarize, our key contributions in this work are as follows: Firstly, to address the lack of reasoning and application-oriented tasks in existing chemical benchmarks, we propose ChemCoTBench,

which evaluates the chemical capabilities of reasoning-LLMs through step-by-step tasks centered on molecular structure modification. Secondly, ChemCoTDataset is provided by ChemCoTBench to facilitate LLMs on chemical reasoning. Finally, extensive experiments demonstrate the effectiveness of ChemCoTBench and its corresponding ChemCoTDataset.

## 2  Related Works

**LLM Chain-of-Thoughts.**    LLMs have progressed from text generators to reasoning systems, with [67]'s Chain-of-Thought enabling stepwise problem decomposition via "slow-thinking" paradigms. These reasoning-enhanced LLMs have shown impressive performance in domains requiring systematic problem-solving skills, particularly in mathematics [51], coding [27], and multi-modality tasks [69]. Models like DeepSeek-R1 [13], Gemini [59], and Anthropic Claude [56] have achieved notable results on mathematical benchmarks like MATH [19] and GSM8K [6], while also excelling at programming. Recent studies have begun exploring LLMs for chemical tasks, such as synthesis planning [4] and computational chemistry [26, 48, 54]. However, these efforts lack a systematic evaluation of LLMs' chemical reasoning capabilities, spanning spatial reasoning, domain-specific knowledge integration, and multi-step logical inference.

**Chemical Benchmarks.**    Current chemical benchmarks primarily focus on assessing discrete knowledge retrieval or simple prediction tasks, rather than evaluating the step-by-step reasoning processes crucial for complex chemical problem-solving. Most existing benchmarks [39, 40, 53, 66] concentrate on question-answering formats that test factual recall and precise calculation but offer limited insight into a model's ability to reason through multi-step chemical problems. Studies like [3, 15, 45] have begun exploring LLMs' chemical capabilities but typically focus on isolated tasks rather than comprehensive reasoning scenarios. Recent work by [73] introduces ChemLLM, a chemistry-specialized LLM framework with supporting datasets, but its benchmark focuses on knowledge recall rather than complex reasoning. Similarly, [15] introduces MolPuzzle, a benchmark for molecular structure elucidation that advances spatial reasoning evaluation but remains limited to spectral interpretation rather than broader chemical reasoning. ChemCoTBench advances chemical reasoning evaluation by using molecular structure to guide step-by-step reasoning, featuring core chemical arithmetic tasks and advanced cross-context applications for more thorough LLM assessment.

## 3  ChemCoTBench Construction

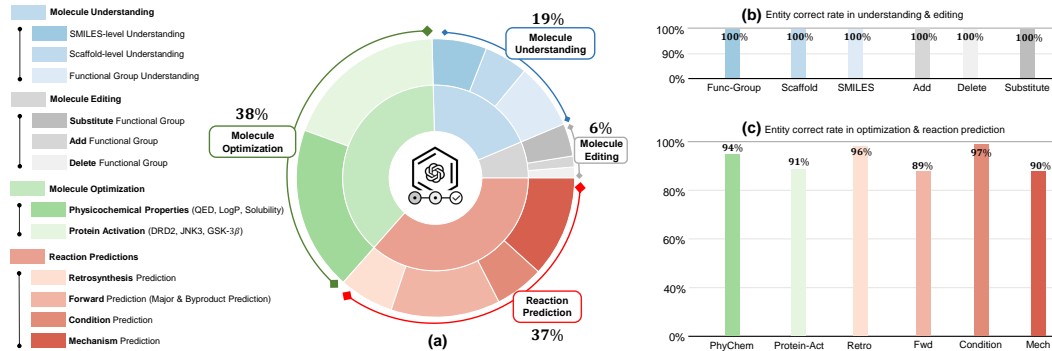

Figure 2: **(a).** Distribution analysis for ChemCoTBench. **(b).** Samples from both molecular understanding and editing tasks achieved exceptionally high accuracy in chemical expert evaluations of chemical entities, including function group names, molecule names, chemical operation names, reaction information, etc. **(c).** Samples from molecule optimization and reaction prediction also show high accuracy (> 89%) in chemical expert evaluations.

ChemCoTBench contains 1,495 samples across 22 chemical tasks as the benchmark dataset, as shown in Fig 2(a). 22,000 high-quality samples with chain-of-thoughts annotations are further sampled to form the ChemCoTDataset. ChemCoTBench was constructed through over 1,800 hours of combined expert and LLM-assisted annotation. It comprises four main tasks and 22 subtasks, covering a broad spectrum of chemical challenges. We define the reasoning steps of each task as modular chemical operations, as shown in the bottom two lines of Fig. 3. ChemCoTBench is guided

by two core principles: **Diversity** and **Quality**. Molecular diversity is ensured by systematically selecting compounds with varied scaffolds and functional groups, enabling broad coverage of real-world chemical scenarios. To ensure high data quality, all benchmark samples undergo multi-stage hybrid review by LLMs and expert chemists, with prompt templates iteratively refined to meet subtask-specific requirements.

## 3.1 Task Construction

To evaluate the capabilities of LLMs in chemistry, we constructed a comprehensive suite of tasks.

**Foundation Task: Molecule-Understanding.** We begin with the recognition and counting of two fundamental elements of molecules: (1) *Functional groups (FGs)*, which are critical clusters of atoms that determine the physicochemical properties and reactivity of organic molecules; (2) *Rings*, which maintain fixed conformations and serve as stable building blocks in drug design, crystal engineering, and polymer synthesis. The recognition and counting of FGs and rings, which require syntactic and lexical understanding of SMILES, remain challenging for LLMs due to their limited chemical topology awareness. Next, we evaluate the recognition of two more complex scaffolds: (1) *Murcko scaffolds*, which are molecular frameworks obtained by systematically removing side chains and serve as a foundation for structural analysis in medicinal chemistry; (2) *Ring systems*, which include fused and bridged ring systems and pose a significant challenge for molecular synthesis. These tasks assess deeper hierarchical comprehension. Finally, we introduce SMILES equivalence tasks, involving permutations and mutations, to test whether LLMs can recognize chemically equivalent structures despite surface-level variations. This probes the models' robustness to SMILES variability.

**Foundation Task: Molecule-Editing.** This task assesses whether LLMs can perform basic molecular editing operations, such as adding, deleting, and substituting functional groups, when guided by natural language instructions. Analogous to basic arithmetic in mathematics, these editing operations form the building blocks of molecular manipulation. Complex tasks like molecular optimization or synthesis can be translated into specific editing operations. For example, a molecular optimization task can be treated as a series of molecule-editing tasks aimed at improving chemical or biological properties. This task evaluates two core capabilities: the capacity to maintain chemical validity after editing operations and the ability to correctly execute the modifications based on textual instructions.

**Application Task: Molecule-Optimization.** This task evaluates whether LLMs can generate optimized molecules given a source molecule and target property. We consider two levels of molecular properties: At the *physicochemical level*, we aim to improve LogP, solubility, and QED for improved drug-likeness. At the *target level*, we aim to improve binding affinity for the DRD2, GSK3-$\beta$, and JNK3 target, which poses a more challenging task as it requires the understanding of drug-target interactions. Solving these problems necessitates in-depth analysis and reasoning capabilities, as LLMs must not only parse the molecular structure but also infer how specific structural modifications influence target properties through complex chemical and biological interactions.

**Application Task: Reaction Prediction.** This task evaluates LLMs' chemical reasoning ability across four tasks: (1) *Forward Prediction*: Predict major products and by-products from reactants and reagents, requiring knowledge of reactivity, reaction rules, and stability. By-product prediction aids reaction optimization and purification by reflecting kinetics and thermodynamics. (2) *Single-Step Retrosynthesis*: Given a product and reagents, predict reactants by identifying key bond disconnections and functional group transformations under constraints. (3) *Reaction Condition Recommendation*: Suggest catalysts, solvents, and reagents for given reactants and products, relying on understanding of solvent effects, catalyst mechanisms, and their impact on yield and selectivity. (4) *Reaction Mechanism Understanding*: Includes Next Elementary-Step Product Prediction (predicting intermediates stepwise, testing electron flow modeling) and Mechanism Route Selection (choosing the most plausible pathway from alternatives, assessing mechanistic reasoning). Together, these tasks span from overall product prediction to detailed mechanistic insight, providing a comprehensive test of LLMs as chemical reasoning agents.

## 3.2 Benchmark Construction

**Data Collection.** Raw molecular structures for understanding, editing, and optimization are sourced from published datasets, including PubChem [31], ChEMBL [11], ZINC [25], and Deep-Mol-Opt [18]. Chemical reactions are collected from patent databases such as USPTO [21], Pistachio [44],

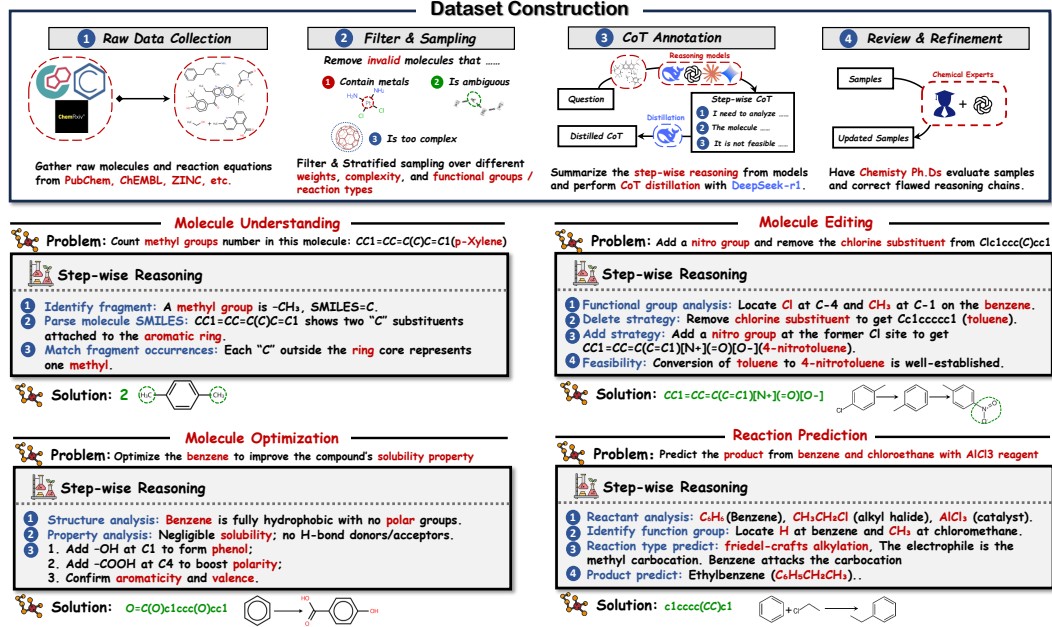

Figure 3: The dataset construction pipeline of ChemCoTBench contains four steps, including raw data collection, molecule filtering and sampling, chain-of-thoughts annotation, and chemical expert review & refinement. We also visualize the samples from the four main tasks and their corresponding modular chemical operations during the reasoning process.

and Reaxys [8]. For reaction mechanism annotation, we refer to the processing pipeline proposed in [29]. The complete data collection protocols are archived in the Appendix B.

**Data Filtering and Sampling.** An initial filtration step removed specimens exhibiting: metal-containing compounds, excessive molecular complexity (defined by the presence of multiple sophisticated functional groups and polycyclic architectures), and factually inconsistent data. To ensure both high data diversity and broad coverage, we systematically curate diverse chemical features across tasks. For molecular understanding, the dataset includes 38 functional groups and 9 ring types. For editing, we cover 57 functional group transformations. Optimization tasks span 4 molecular weight-based structural scales. For reaction tasks, we include 100 common reaction classes, 175 distinct reaction conditions, and 123 annotated reaction mechanisms. Together, these components offer a rich and representative benchmark dataset for evaluating chemical reasoning in LLMs.

**Chain-of-Thoughts Annotation for Modular Chemical Operations.** To derive intermediate reasoning steps for complex chemical problems, we distill the chain-of-thought annotations from LLMs and arrange them as modular chemical operations for systematic evaluation and supervised fine-tuning of reasoning models. Specifically, we analyze the problem-solving strategies of state-of-the-art reasoning models, including Gemini-2.5-pro, DeepSeek-R1, and Claude-3.7-sonnet-thinking, to extract step-wise reasoning patterns. These are distilled into a structured training corpus using DeepSeek-R1 via CoT prompting. As illustrated in Fig. 3, our distilled CoT samples span key chemical tasks including molecular understanding, editing, optimization, and reaction prediction.

### 3.3 Quality Review & Refinement

To ensure the high quality of our benchmark and its large-scale dataset, we performed iterative evaluation and optimization of the molecules, results, and distilled Chain-of-Though reasoning processes from DeepSeek-R1-0324 [13] in ChemCoTBench. Our hybrid assessment approach combines automated LLM-based evaluation for scalability with manual expert review by chemists to guarantee scientific rigor, enabling comprehensive dataset refinement while maintaining efficiency.

**LLM-based CoT Evaluation.** To improve the quality of CoT annotations in Deepseek's process, we focused on two key elements: (1) *Task-Specific Prompt Design*: We discovered that providing detailed task descriptions and prior knowledge within prompts significantly enhances the model's performance on chemical tasks. (2) *Incorporation of IUPAC name*: We found that including IUPAC

Table 1: Experiments for the foundational tasks, including molecule understanding, molecule editing, and their correlated subtasks. For the functional-group counting task (FG) and ring counting task (Ring) in the functional-group level molecule understanding, we apply the mean absolute error (MAE) as the evaluation metric. Tanimoto molecule similarity is applied as the evaluation for the Murcko scaffold extraction task (Murcko). The accuracy (%) metric is applied to other subtasks.

| Models | Func-Group | | Scaffold | | SMILES | Molecule-Edit | | |
| --- | --- | --- | --- | --- | --- | --- | --- | --- |
| | FG↓ | Ring↓ | Murcko↑ | Ring-sys↑ | Eq.↑ | Add | Delete | Sub |
| *W/ Thinking* | | | | | | | | |
| Gemini-2.5-pro-think | **0.11** | **0.60** | **0.51** | **87.5** | 82 | **100** | **85** | 81.7 |
| Claude3.7-sonnet-think | 0.21 | 1.60 | 0.40 | 80.0 | **84** | 85 | 80 | **83.4** |
| DeepSeek-R1 | 0.27 | 1.55 | 0.34 | 45.0 | 65 | 70 | 70 | 68.3 |
| o3-mini@20250103 | 0.13 | **0.60** | 0.39 | 75.0 | 78 | 65 | 55 | 80.0 |
| o1-mini@20240912 | 0.21 | 1.25 | 0.25 | 61.7 | 66 | 55 | 80 | 58.3 |
| Qwen3-235B-A22B-think | 0.42 | 1.00 | 0.38 | 82.5 | 72 | 40 | 75 | 71.7 |
| Qwen3-32B-think | 0.25 | 0.95 | 0.21 | 75.0 | 68 | 20 | 55 | 20.0 |
| Llama-Nemo-49B-think | 0.80 | 1.90 | 0.09 | 86.8 | 46 | 0 | 80 | 8.0 |
| *W/o Thinking* | | | | | | | | |
| GPT-4o@20241120 | 0.17 | 1.35 | 0.21 | 80.0 | 72 | 80 | 80 | 65.0 |
| Deepseek-V3 | 0.15 | 1.50 | 0.24 | 76.7 | 77 | 70 | 75 | 76.7 |
| Gemini-2.0-flash | 0.19 | 1.65 | 0.43 | 75.0 | 76 | 65 | 75 | 66.7 |
| Qwen3-235B-A22B | 0.42 | 1.00 | 0.34 | 82.5 | 75 | 40 | 75 | 66.7 |
| Qwen3-32B | 0.26 | 0.95 | 0.22 | 68.3 | 67 | 30 | 55 | 25.0 |
| Qwen2.5-72B-Instruct | 0.26 | **0.60** | 0.24 | 70.0 | 61 | 70 | 80 | 56.7 |
| Qwen2.5-32B-Instruct | 0.36 | 0.65 | 0.12 | 53.3 | 62 | 50 | 50 | 48.3 |
| Llama-3.1-70B-Instruct | 0.52 | 1.80 | 0.12 | 68.3 | 67 | 60 | 80 | 50.0 |
| Llama-Nemo-49B | 0.72 | 1.77 | 0.11 | 65.0 | 54 | 30 | 55 | 30.5 |
| Gemma-2-27b-it | 0.19 | 1.65 | 0.43 | 66.7 | 76 | 75 | 70 | 35.0 |
| Phi-4-14B | 0.28 | 1.65 | 0.15 | 70.0 | 65 | 60 | 80 | 38.3 |
| OLMo2-32B-Instruct | 0.19 | 1.05 | 0.07 | 63.3 | 50 | 15 | 30 | 11.7 |
| *Domain Expert Models* | | | | | | | | |
| Ether0 | Failed | 0.35 | Failed | Failed | 63 | 94 | 76 | 78 |
| BioMedGPT-7B | 1.6 | 2.43 | 0.18 | 53.3 | 39 | 10 | 12 | 10 |
| BioMistral-7B | 1.0 | 1.85 | 0.04 | 32.5 | 50 | 0 | 10 | 0 |

names helps LLMs better understand complex molecular structures, as these names offer precise details about functional groups. Leveraging these insights, we iteratively refined our prompt designs. We then employed GPT-4o as an LLM verifier to ensure each CoT annotation was consistent with its corresponding prompt template and the provided IUPAC names.

**Chemical Expert Review & Refinement**   As a rigorous benchmark evaluation, we engaged 13 chemistry PhD candidates from Top Universities to assess the accuracy of chemical entities, including functional groups, molecular names, reaction types, and operation names, in ChemCoTBench's CoT annotations. As shown in Fig. 2 (b), the evaluation revealed near-perfect accuracy for molecule understanding and editing tasks, while more challenging tasks like molecule optimization and reaction prediction maintained over 90% accuracy (as shown in Fig. 2 (c)). Furthermore, we corrected these errors to enhance ChemCoTBench's quality.

## 4   Experiments

### 4.1   Evaluation Metrics

For understanding tasks, functional group (FG) and ring recognition are treated as counting problems, with mean absolute error (MAE) used to measure precision. Scaffold-level understanding includes extracting Murcko scaffolds, evaluated by Tanimoto similarity, and identifying whether complex ring systems are present, evaluated by accuracy. The SMILES equivalence task is formulated as a binary decision problem, determining whether the target and source SMILES represent the same molecule, and is also evaluated using accuracy. For molecule editing, we use Pass@1 to assess whether the edited molecule meets the instructions. Mechanism route selection is framed as a multiple-choice task

Table 2: Baseline Performance on Molecule Optimization. The optimized targets are categorized into physicochemical properties (QED, LogP, solubility) and protein activity-related properties (JNK3, DRD2, GSK-3$\beta$), with the latter posing greater challenges to the model's chemical knowledge and reasoning capabilities. $\Delta$ is the mean property improvement, where a negative $\Delta$ indicates that most optimizations are property degradations. SR% is the success rate that brings property increase.

| Models | LogP | | Solubility | | QED | | DRD2 | | JNK3 | | GSK3-$\beta$ | |
|---|---|---|---|---|---|---|---|---|---|---|---|---|
| | $\Delta$ | SR% | $\Delta$ | SR% | $\Delta$ | SR% | $\Delta$ | SR% | $\Delta$ | SR% | $\Delta$ | SR% |
| *W/ Thinking* | | | | | | | | | | | | |
| Gemini-2.5-pro-think | -0.22 | 76 | 1.06 | 70 | 0.28 | 84 | 0.36 | **74** | -0.02 | 35 | 0.06 | **68** |
| Claude3.7-sonnet-think | 0.41 | **80** | 0.37 | 75 | 0.12 | 73 | 0.17 | 63 | 0.01 | **49** | 0.02 | 57 |
| DeepSeek-R1 | 0.47 | 69 | 0.80 | 80 | 0.17 | 72 | 0.12 | 62 | -0.02 | 29 | 0.01 | 41 |
| o3-mini@20250103 | 0.26 | 59 | 0.81 | 85 | 0.21 | **86** | 0.19 | 69 | -0.03 | 23 | 0.01 | 45 |
| o1-mini@20240912 | -0.42 | 52 | 1.78 | **95** | 0.07 | 70 | -0.03 | 37 | -0.10 | 15 | -0.08 | 31 |
| Qwen3-235B-A22B-think | 0.05 | 40 | 0.20 | 40 | 0.02 | 24 | 0.03 | 31 | -0.01 | 23 | 0.01 | 31 |
| Qwen3-32B-think | -0.01 | 1 | 0.13 | 19 | 0.01 | 9 | 0.0 | 4 | -0.02 | 3 | -0.02 | 6 |
| Llama-Nemo-49B-think | -0.24 | 7 | 0.25 | 25.2 | 0.10 | 41 | 0.03 | 29.9 | -0.02 | 6 | -0.01 | 11.2 |
| *W/o Thinking* | | | | | | | | | | | | |
| GPT-4o@20241120 | -0.09 | 37 | 0.92 | 80 | 0.13 | 70 | 0.07 | 48 | -0.02 | 30 | -0.00 | 39 |
| DeepSeek-V3 | 0.09 | 33 | 0.57 | 92 | 0.08 | 46 | 0.03 | 28 | 0.00 | 18 | -0.01 | 29 |
| Gemini-2.0-flash | 0.37 | 72 | 0.28 | 58 | 0.13 | 79 | 0.15 | 63 | -0.02 | 34 | 0.01 | 38 |
| Qwen3-235B-A22B | 0.03 | 21 | 0.18 | 45 | 0.07 | 34 | 0.04 | 26 | -0.01 | 18 | 0.02 | 25 |
| Qwen3-32B | 0.0 | 2 | 0.08 | 20 | 0.02 | 14 | -0.01 | 6 | -0.02 | 6 | -0.02 | 5 |
| Qwen2.5-72B-Instruct | -0.03 | 41 | 0.34 | 59 | 0.07 | 57 | 0.04 | 40 | -0.02 | 26 | -0.00 | 40 |
| Qwen2.5-32B-Instruct | 0.15 | 44 | 0.49 | 65 | 0.09 | 54 | 0.05 | 32 | -0.02 | 19 | 0.01 | 31 |
| Llama-3.1-70B-Instruct | 0.02 | 35 | 0.72 | 81 | 0.15 | 61 | -0.00 | 31 | -0.01 | 30 | 0.01 | 40 |
| Llama-Nemo-Super-49B | -0.01 | 24 | 0.34 | 40 | 0.08 | 43 | -0.00 | 16 | -0.00 | 15 | 0.01 | 27 |
| Gemma-2-27b-it | 0.01 | 31 | 0.39 | 69 | 0.07 | 56 | -0.02 | 15 | -0.00 | 16 | -0.00 | 17 |
| Phi-4-14B | -0.26 | 44 | 0.22 | 53 | 0.17 | 74 | -0.02 | 18 | -0.03 | 14 | -0.00 | 22 |
| OLMo2-32B-Instruct | -1.71 | 11 | 1.21 | 46 | 0.08 | 40 | -0.05 | 7 | -0.03 | 8 | -0.02 | 12 |
| *Domain Expert Models* | | | | | | | | | | | | |
| Ether0 | 0.0 | 0 | 0.0 | 0 | 0.0 | 0 | 0.0 | 0 | 0.0 | 0 | 0.0 | 0 |
| BioMedGPT-7B | -0.36 | 17 | 0.25 | 63 | -0.29 | 7 | -0.09 | 5 | -0.11 | 6 | -0.08 | 1 |
| BioMistral-7B | 0.01 | 1 | 0.24 | 6 | 0.0 | 0 | 0.0 | 1 | -0.01 | 1 | -0.01 | 0 |

and evaluated by accuracy. Other reaction tasks are modeled as SMILES generation problems, where evaluation is based on both Top-1 accuracy and fingerprint-based similarity (FTS), using Morgan [49], MACCS [7], and RDKit [33] fingerprints to reflect correctness and structural similarity.

## 4.2 Evaluated LLMs

Our evaluation includes three model categories: (1) **Reasoning LLMs** with explicit step-by-step reasoning, including Deepseek-R1 [13], o1-mini [61], o3-mini [62], Gemini-2.5-pro [58], Claude-3.7-Sonnet-thinking [56], Qwen-3-thinking [63], Llama-Nemotron-thinking [2]; (2) **General-purpose non-reasoning LLMs** without specialized reasoning mechanisms including GPT-4o [24], Qwen-2.5/3 [70], Llama-3.3 [12], Gemma-2 [60], Phi-4 [1], OLMo2 [47] (3) **Biomolecular LLMs** BioMedGPT [41], BioMistral [32], and Text+Chem T5 [5]. This comprehensive comparison evaluates whether reasoning-specific capabilities provide advantages over domain-specific models in challenging chemical reasoning tasks. Details of evaluation implementation in Appendix C.2.

## 4.3 LLMs' Performance on Solving ChemCoTBench

We evaluated reasoning LLMs, their non-reasoning counterparts, and task-specific models [5, 32, 41] on foundational (molecule understanding and editing, Table 1) and application (molecule optimization, Table 2; reaction prediction, Table 3) tasks within ChemCoTBench. Key findings include:

**Hierarchical Skill Transfer.** Strong performance in foundational molecular understanding and editing tasks directly translates to success in complex application tasks. This validates ChemCoTBench's design, where fundamental chemical knowledge underpins advanced problem-solving. For example,

Table 3: The chemical reaction task contains forward prediction (Fwd$_{major}$: major-product prediction, and Fwd$_{by}$: by-product prediction), resynthesis prediction (Retro), reaction condition prediction (Condition), and reaction mechanism prediction (NEPP: next element-step product prediction, MechSel: reaction mechanism selection prediction). FTS: molecule fingerprint similarity with reference.

| Models | Fwd $_{major}$ | | Fwd $_{by}$ | | Retro | | Condition | | NEPP | | MechSel |
|---|---|---|---|---|---|---|---|---|---|---|---|
| | Top-1 | FTS↑ | Top-1 | FTS↑ | Top-1 | FTS↑ | Top-1 | FTS↑ | Top-1 | FTS↑ | Acc.↑ |
| *W/ Thinking* | | | | | | | | | | | |
| Gemini-2.5-pro-think | 0.72 | **0.89** | 0.20 | **0.51** | **0.20** | **0.45** | 0.20 | **0.33** | **0.58** | 0.53 | **0.62** |
| Claude3.7-sonnet-think | **0.73** | 0.87 | **0.25** | 0.31 | 0.12 | 0.27 | 0.14 | 0.22 | 0.24 | **0.79** | 0.49 |
| DeepSeek-R1 | 0.48 | 0.71 | 0.21 | 0.45 | 0.07 | 0.41 | **0.23** | 0.30 | 0.15 | 0.55 | 0.46 |
| o3-mini@20250103 | 0.52 | 0.71 | 0.20 | 0.27 | 0.11 | 0.39 | 0.19 | 0.19 | 0.18 | 0.58 | 0.49 |
| o1-mini@20240912 | 0.26 | 0.31 | 0.11 | 0.17 | 0.02 | 0.15 | 0.08 | 0.22 | 0.09 | 0.33 | 0.44 |
| Qwen3-235B-A22B-think | 0.03 | 0.54 | 0.0 | 0.07 | 0.01 | 0.42 | 0.20 | 0.27 | 0.09 | 0.63 | 0.41 |
| Qwen3-32B-think | 0.11 | 0.33 | 0.09 | 0.18 | 0.02 | 0.24 | 0.14 | 0.20 | 0.08 | 0.67 | 0.46 |
| Llama-Nemo-49B-think | 0.09 | 0.18 | 0.04 | 0.18 | 0.0 | 0.05 | 0.18 | 0.19 | 0.04 | 0.21 | 0.47 |
| *W/o Thinking* | | | | | | | | | | | |
| GPT-4o@20241120 | 0.28 | 0.58 | 0.04 | 0.20 | 0.03 | 0.43 | 0.0 | 0.08 | 0.12 | 0.71 | 0.43 |
| DeepSeek-V3 | 0.36 | 0.62 | 0.04 | 0.30 | 0.03 | 0.44 | 0.08 | 0.16 | 0.20 | 0.70 | 0.45 |
| Gemini-2.0-flash | 0.19 | 0.56 | 0.01 | 0.07 | 0.05 | 0.41 | 0.07 | 0.08 | 0.13 | 0.68 | 0.53 |
| Qwen3-235B-A22B | 0.04 | 0.57 | 0.0 | 0.06 | 0.0 | 0.30 | 0.07 | 0.14 | 0.07 | 0.59 | 0.40 |
| Qwen3-32B | 0.06 | 0.57 | 0.0 | 0.13 | 0.0 | 0.43 | 0.01 | 0.10 | 0.08 | 0.67 | 0.46 |
| Qwen2.5-72B-Instruct | 0.04 | 0.49 | 0.0 | 0.13 | 0.01 | 0.35 | 0.01 | 0.07 | 0.06 | 0.60 | 0.46 |
| Qwen2.5-32B-Instruct | 0.01 | 0.43 | 0.0 | 0.12 | 0.0 | 0.29 | 0.02 | 0.10 | 0.05 | 0.50 | 0.45 |
| Llama-3.1-70B-Instruct | 0.02 | 0.35 | 0.0 | 0.08 | 0.0 | 0.34 | 0.06 | 0.13 | 0.06 | 0.41 | 0.39 |
| Llama-Nemo-49B | 0.04 | 0.40 | 0.0 | 0.08 | 0.0 | 0.30 | 0.03 | 0.05 | 0.05 | 0.41 | 0.46 |
| Gemma-2-27b-it | 0.01 | 0.55 | 0.0 | 0.04 | 0.0 | 0.48 | 0.03 | 0.10 | 0.04 | 0.53 | 0.43 |
| Phi-4-14B | 0.01 | 0.27 | 0.03 | 0.10 | 0.0 | 0.39 | 0.0 | 0.03 | 0.05 | 0.57 | 0.39 |
| OLMo2-32B-Instruct | 0.0 | 0.10 | 0.0 | 0.07 | 0.0 | 0.10 | 0.0 | 0.03 | 0.01 | 0.13 | 0.32 |
| Text+Chem T5 | 0.44 | 0.74 | 0.0 | 0.07 | 0.06 | 0.24 | 0.0 | 0.09 | 0.0 | 0.0 | 0.10 |

Claude-3.7-sonnet and Gemini-2.5-pro, top performers in foundational tasks (Table 1), also lead in molecule optimization and reaction prediction.

**Efficacy of Advanced Reasoning in Commercial LLMs:** Commercial LLMs equipped with sophisticated reasoning mechanisms (e.g., Deepseek-R1, o3-mini) significantly outperform their non-reasoning counterparts on ChemCoTBench's challenging applied tasks. In molecule optimization (Table 2), Deepseek-R1 shows a >30% improvement over Deepseek-V3, and o3-mini gains >20% over GPT-4o. Similar trends are observed for reaction prediction (Table 3). This suggests that RL-honed "slow thinking" capabilities [42, 51, 65], when combined with sufficient domain knowledge, enable superior abstraction and problem-solving beyond mere knowledge retrieval.

**Unrealized Promise of Hybrid Thinking in Open-Source Models for Chemistry without Domain-Specific Data:** Current open-source models featuring hybrid thinking modes, such as Llama-3.3-Nemotron [2] and Qwen3 [76], achieve substantial, often efficient, performance in general domains like code and mathematics. However, their advanced reasoning capabilities, intended to be general, do not effectively transfer to specialized scientific fields like chemistry. We attribute this shortfall to a critical lack of domain-specific reasoning training data. Our empirical results are stark (Tables 1-3): enabling the reasoning modes in these models yields no significant performance improvement on chemical tasks compared to their non-reasoning counterparts. This finding strongly suggests that general reasoning architectures require specialized data to adapt to new domains.

### 4.4 Evaluating Distillation Strategies in Chemical Reasoning

Our preceding analyses underscored the critical role of advanced reasoning capabilities (or "slow thinking") for tackling complex chemical tasks. This motivates our exploration of distillation strategy [13] as a standard method to bolster this capability in open-source LLMs.

**Challenges in Distilling Chemical Reasoning:** Distilling CoT capabilities from advanced LLMs (e.g., using DeepSeek-R1-generated samples [13, 75]) is a common strategy to enhance reasoning

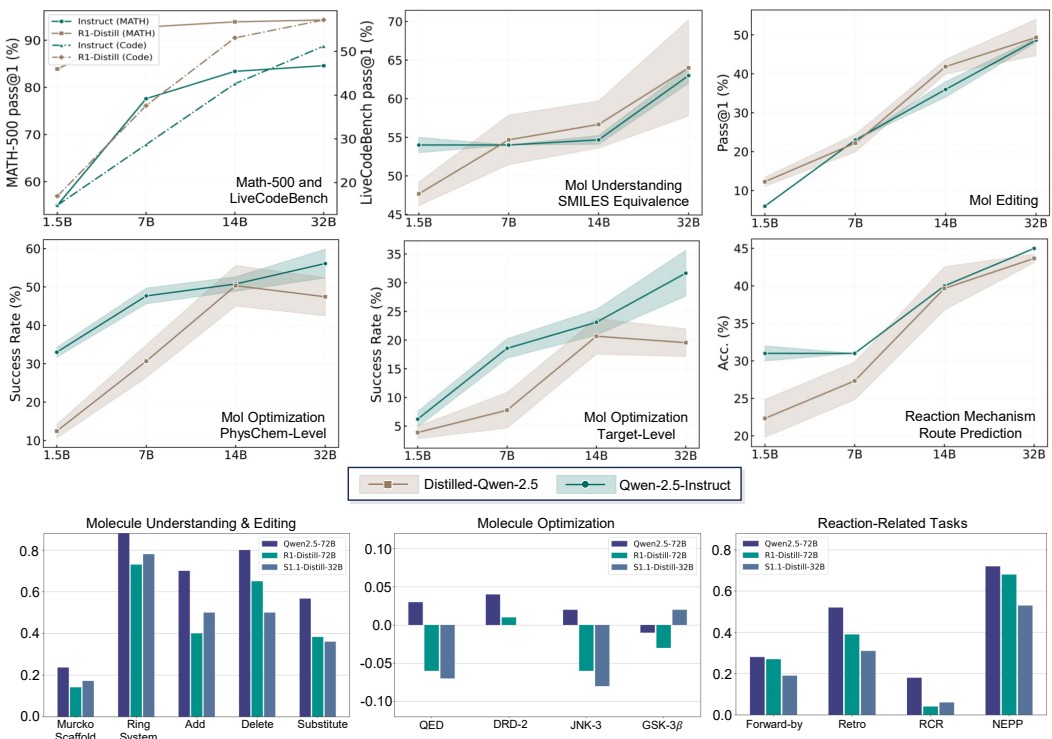

Figure 4: The top two rows compare the reasoning performance of the Qwen-2.5-Instruct series against its DeepSeek-R1-distilled versions. The bottom row propose the comparison between two reasoning models with distillation strategy (R1-Distill, S1.1-Distill), and their Qwen-2.5 backbone.

in smaller models. However, this approach proves significantly limited for specialized chemical reasoning. Our experiments (Fig.4) show that Qwen2.5-Instruct models distilled for CoT exhibit little to no improvement on ChemCoTBench chemical subtasks compared to their non-distilled counterparts; indeed, smaller base models (1.5B-32B) often perform comparably or better. While effective for general domains like code and math (Fig.4), this distillation strategy falters in chemistry, likely due to insufficient volume or specificity of chemical CoT samples in the distillation process, hindering the development of robust step-by-step chemical reasoning. Moreover, smaller distilled models (<7B) frequently produce lengthy, repetitive, and irrelevant (hallucinatory) thought processes. These findings suggest that direct CoT distillation, without substantial domain-specific adaptation, is an ineffective standalone method for improving chemical reasoning in open-source models. From the bottom row of Fig.4, an inverse correlation is observed between the model's performance on Math/Code and its OOD performance in chemistry: specifically, S1.1-distill [46] outperforms R1-distill [13] on MATH500 but underperforms it on multiple chemical subtasks.

## 4.5 Effective Methods for Enhancing Chemical Reasoning

Given the limitations of direct distillation, we explore effective strategies to enhance chemical reasoning capabilities. We investigate two approaches: prompting with chemical reasoning templates and supervised fine-tuning (SFT) with our ChemCoTDataset.

**Prompt engineering cannot bring stable chemical reasoning improvements:** we first evaluate the CoT prompting strategy: providing only coarse strategic guidance (CoT templates). The results in Table 4 demonstrate that the prompting strategy cannot yield stable and significant performance gains across all chemical reasoning tasks. Specifically, for the functional-group detection task, models from 1.5B to 32B show stable improvements. However, for other chemical reasoning tasks, CoT prompting strategy shows unstable influence due to the lack of chemical knowledge.

**SFT on ChemCoTDataset boosts chemical reasoning:** We further explored enhancing chemical reasoning using our high-quality, domain-specific ChemCoTDataset via supervised funetuning. This dataset was meticulously curated to minimize hallucinations and align with expert thought processes,

Table 4: Investigation of methods to enhance chemical reasoning. We propose two approaches: prompting with chemical reasoning templates and supervised fine-tuning with ChemCoTDataset. Experimental results with Qwen-2.5 backbones (different scales from 1.5B to 32B) demonstrate that coarse guidance from reasoning templates cannot yield stable performance, while SFT with Chem-CoTDataset provides significant reasoning boosting, verifying the effectiveness of ChemCoTDataset.

| Models | Mol-Understanding | | | | Mol-Editing | | |
|---|---|---|---|---|---|---|---|
| | Func-Group↓ | Ring↑ | Murcko↑ | Ring-System↑ | Add↑ | Delete↑ | Substitute↑ |
| 1.5B | 1.32 | 1.17 | 0.07 | 0.15 | 0 | 0.15 | 0.05 |
| 1.5B-CoT-Template | 0.40 | 1.04 | 0.07 | 0.58 | 0.05 | 0.15 | 0.02 |
| 1.5B-CoT-SFT | **0.35** | 0.69 | **0.12** | **0.78** | **0.20** | **0.25** | **0.07** |
| 7B | 0.43 | 1.04 | 0.09 | 0.82 | 0.15 | 0.3 | 0.15 |
| 7B-CoT-Template | 0.25 | 1.21 | 0.09 | 0.57 | 0.15 | 0.45 | 0.15 |
| 7B-CoT-SFT | 0.33 | 0.69 | **0.31** | 0.45 | **0.40** | **0.45** | **0.15** |
| 14B | 0.42 | 1.1 | 0.11 | 0.67 | 0.35 | 0.65 | 0.2 |
| 14B-CoT-Template | 0.35 | 0.91 | 0.12 | 0.62 | 0.3 | 0.4 | 0.2 |
| 14B-CoT-SFT | 0.41 | 0.70 | **0.25** | 0.63 | **0.35** | **0.70** | **0.38** |
| 32B | 0.35 | 0.95 | 0.15 | 0.60 | 0.45 | 0.55 | 0.5 |
| 32B-CoT-Template | 0.33 | 0.74 | 0.12 | 0.70 | 0.4 | 0.65 | 0.4 |
| 32B-CoT-SFT | **0.29** | 0.72 | **0.17** | **0.72** | **0.55** | **0.66** | **0.53** |

which we posited would be vital for chemical reasoning tasks. We test this by evaluating the SFT strategy augmented with detailed step-by-step reasoning processes from our dataset. The results in Table. 4 consistently demonstrate that our large-scale chemical CoT dataset significantly enhances the chemical reasoning capabilities of Qwen-2.5 models across various scales (1.5B to 32B) when used in this way. Augmentation with SFT processes yielded stable and substantial performance gains across all evaluated tasks.

## 5   Conclusion and Discussion

This paper introduces ChemCoTBench, a new chemical reasoning benchmark to evaluate the complex chemical problem-solving ability of LLMs. Compared to existing Scientific benchmarks that focus on simple knowledge retrieval, our ChemCoTBench establishes a *step-by-step, application-oriented, and high-quality* benchmark by gathering samples from both foundational and applicational chemical tasks, including molecule understanding, editing, optimization, and reaction prediction. Furthermore, a 22k large chemical CoT dataset, ChemCoTDataset, is also provided for enhancing chemical reasoning ability of LLMs. Extensive experiments across 22 chemical tasks in ChemCoTBench demonstrate that current open-source and distillation-based reasoning LLMs still have significant room for improvement in complex chemical reasoning, while also validating the boosting effect of our large chemical CoT dataset on chemical reasoning capabilities. ChemCoTBench bridges the gap between LLM reasoning capabilities and real-world chemical problem-solving needs, offering researchers a standardized evaluation platform for complex chemical reasoning. Future works could continue with designing policy optimization and distillation strategies to enhance the chemical reasoning capability of LLMs. Chemical-aware reward mechanisms warrant further exploration. We also focus on extending ChemCoTBench and its chemical CoT dataset to larger biochemical domains and scale.

## Acknowledgements

Hao Li and He Cao are equal contributors. This work was supported in part by the Natural Science Foundation of China (No. 62202014, 62332002, 62425101) and Shenzhen Hetao Shenzhen-Hong Kong Science and Technology Innovation Cooperation Zone, under Grant No. HTHZQSWS-KCCYB-2023052.

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

# Appendix

## A  Full Related Works

### A.1  LLM Chain-of-Thoughts.

The evolution of large language models (LLMs) has transitioned from basic text generation to sophisticated reasoning systems, exemplified by [67] Chain-of-Thought methodology, which facilitates systematic problem decomposition through deliberate cognitive paradigms. These advanced reasoning architectures demonstrate exceptional proficiency in domains that demand structured analytical capabilities, particularly in mathematical computations and programming tasks. Benchmark evaluations on MATH [19] and GSM8K [6] reveal significant achievements by models including DeepSeek-R1 [13], Gemini [59], and Anthropic Claude.

**LLM Reasoning on Multimodal Domain.**  With the rapid development of the vision-language domain, reasoning on images and videos is increasingly important [68]. Visual-RFT [37], VLM-R1 [52] establish the visual chain-of-thoughts data construction pipeline and RL-based post-training strategies. Vision-R1 [20] further proposes the cold start strategy for better multimodal reasoning. In the 3D domain, [9, 36, 55, 72, 74] apply chain-of-thoughts to point clouds and 3D objects to achieve LLM reasoning.

**LLM Reasoning on Chemical Domain.**  Emerging applications in chemical sciences demonstrate LLM capabilities in spectra analysis [34], synthesis planning [4], and computational chemistry [26, 48, 54]. Also, LLMs [43] show outstanding multi-task generalization ability on the molecule domain and protein domain. However, current research lacks a comprehensive assessment of chemical reasoning capacities encompassing spatial cognition, domain knowledge assimilation, and complex logical inference processes.

## B  Data Construction Details

In this section, we propose the detailed information during our benchmark and dataset construction process, including the data source description, dataset composition, filtering strategies, and the

Table 5: **The Dataset Statistics of ChemCoTBench and its Large CoT Dataset.** We visualize the sample numbers for every subtask in ChemCoTBench. The data distribution of molecule understanding & editing, molecule optimization, and reaction prediction is nearly average.

| # | Mol-Understanding | | | Mol-Edit | | | Mol-Optimization | | Reaction | | | |
|---|---|---|---|---|---|---|---|---|---|---|---|---|
| | Func-Group | Scaffold | SMILES | Add | Del | Sub | Physico | Protein | Fwd | Retro | Cond | Mech |
| Bench mark | 120 | 100 | 100 | 20 | 20 | 60 | 300 | 300 | 200 | 100 | 90 | 275 |
| CoT Dataset | 6400 | | | 4500 | | | 3183 | 2404 | 2053 | 2165 | 1886 | - |

rationale for dataset construction. In Table. 5, we also visualize the data distribution of subtasks in ChemCoTBench.

## B.1 Data Collection

The raw molecular structures used for understanding, editing, and optimization are obtained from several published datasets, including PubChem [31], ChEMBL [11], ZINC [25], and Deep-Mol-Opt [18]. Chemical reaction data are separately collected from patent databases, including USPTO [21], Pistachio [44], and Reaxys [8]. For reaction mechanism annotation, we followed the processing pipeline described in [29].

## B.2 Dataset Composition and Filtering Strategies

**Molecular Samples (25% of Benchmark):** Although the ZINC database contains 250,000 molecules, we observed that its molecular weight distribution is relatively concentrated. To ensure diversity, we carefully selected molecules from PubChem, ChEMBL, and ZINC based on molecular weight and structural complexity. This filtering process resulted in a smaller but more representative molecular subset for our benchmark.

**Molecular Optimization Pairs (38% of Benchmark):** The Deep-Mol-Opt dataset provided 198,559 molecular pairs with property annotations. However, we excluded pairs with minimal property improvement ($\Delta < 0.3$) or those containing complex polycyclic structures that might challenge LLM comprehension. The remaining high-quality pairs were retained for molecular optimization tasks.

**Chemical Reaction Samples (19% of Benchmark):** Reaction equations (including reactants, products, conditions, and catalysts) were sourced from USPTO, Pistachio, and Reaxys. To avoid redundancy, we balanced the selection across these databases by reaction type and catalyst diversity. For reaction mechanism annotation, we incorporated 275 manually curated examples from [29], which were chosen for their high quality and balanced distribution.

## B.3 Rationale for Task Construction

**Molecular Understanding and Editing Tasks:** Molecular understanding and editing tasks are designed as closed-ended problems with deterministic answers. Since these tasks rely on well-defined chemical properties and structures, we directly sampled molecules from PubChem, ChEMBL, and ZINC as the source data. The corresponding ground-truth answers, including molecular properties and SMILES transformations, are programmatically extracted using RDKit, ensuring accuracy and reproducibility.

**Molecular Optimization Task Design:** Unlike fixed-answer tasks, molecular optimization is inherently open-ended, where multiple valid optimization paths may exist for a given input molecule. To construct this dataset, we considered two sampling strategies:

• Baseline Model-Generated Optimizations: *Advantage*: Enables sampling large-scale and multi-step optimization paths for source molecules; *Limitation*: Existing models often fail to preserve scaffold consistency, a critical requirement in drug design.

- Predefined Molecular Pairs: *Advantage*: Ensures chemically meaningful transformations with verified property improvements; *Limitation*: limited molecule samples.

To maintain the scaffold consistency, we adopt the second strategy for our ChemCoTBench, sourcing molecular pairs from Deep-Mol-Opt [18]. We perform Murcko scaffold similarity analysis to validate scaffold consistency, confirming that the selected pairs maintain structural integrity while optimizing target properties.

**Reaction Prediction Task Design:** Reaction prediction is a cornerstone of chemical research and industrial applications. From an academic standpoint, it is fundamental to understanding chemical reactivity, discovering novel transformations, and advancing the design of new molecules. In practical applications, accurate reaction prediction accelerates drug discovery, facilitates materials science innovation, optimizes chemical manufacturing processes, and enables the automation of chemical synthesis. Our benchmark aims to evaluate LLMs' capabilities in this multifaceted domain rigorously.

- *Forward Reaction Prediction*: This task, pivotal for academic discovery and industrial applications like drug development, evaluates an LLM's ability to predict both major products and, uniquely in our benchmark, byproducts from given reactants and reagents. Data is sourced from 100 distinct reaction classes from Pistachio. To enhance difficulty and assess deeper reasoning, the reaction type is deliberately omitted, requiring the model to first infer the plausible reaction type and then deduce potential products, thereby providing a comprehensive understanding of reaction outcomes crucial for optimization.

- *Retrosynthesis Prediction*: Essential for planning the synthesis of novel compounds, this task assesses an LLM's understanding of reverse chemical logic, specifically its capacity to identify strategic bond disconnections and propose valid precursor structures. We focus on single-step retrosynthesis, considering multi-step planning a more complex hybrid task, to directly evaluate core retrosynthetic reasoning. Data comprises 100 reaction classes from Pistachio, and problem formulation includes providing reagents alongside the target product to help narrow the solution space and guide the LLM towards chemically relevant disconnections.

- *Reaction Condition Prediction*: Predicting optimal reaction conditions (catalysts, solvents, reagents) is critical for synthesis success, efficiency, and selectivity. This task tests an LLM's knowledge of how these components influence reaction pathways. Following Gao et al. [10] for data construction from USPTO [38] (retaining reactions with at most one catalyst, two solvents, and two reagents), we uniquely model this as a SMILES sequence generation task for catalyst, solvent, and reagent prediction, offering a more rigorous challenge than simple MCQ formats by requiring specific chemical structure (In SMILES) generation.

- *Mechanism Prediction*: Understanding reaction mechanisms—the step-by-step sequence of elementary reactions—is fundamental to chemistry, providing the "why" and "how" behind transformations and enabling rational design and optimization. This task evaluates an LLM's grasp of core mechanistic principles such as electron flow, intermediate stability, bond-making/breaking sequences, and the influence of conditions on pathways, addressing a significant gap in current LLM assessments, which often treat reactions as black boxes. Inspired by prior works [28, 29] but aiming for a more holistic probe, we introduce two subtasks: "Next Elementary Step Product Prediction," where the LLM, given a sequence of annotated elementary steps, predicts the subsequent product, testing its ability to comprehend and extrapolate mechanistic progression; and "Reaction Mechanism Selection (MCQ type)," where the LLM chooses the most plausible mechanism from several alternatives for a given reaction (reactants, conditions, reagents), assessing its capacity to discern how subtle changes in reagents or conditions dictate specific mechanistic routes, thereby evaluating both sequential understanding and discriminative judgment of mechanistic pathways.

## C  Experimental Details

### C.1  Hardware Requirements

The experimental workload was supported by a dedicated GPU cluster comprising three high-performance computing nodes: an NVIDIA RTX A6000 (48GB VRAM) and an RTX 3090 (24GB VRAM) for LLM API scheduling and deployment of smaller models (1.5B/7B parameters), complemented by an NVIDIA A100 (80GB VRAM) node dedicated to large-scale LLM inference. This

heterogeneous configuration achieved optimal resource allocation, with the A100's tensor cores and high-bandwidth memory handling memory-intensive model inferences while the A6000/3090 pair efficiently managed concurrent API requests and lighter workloads. Storage requirements remained modest at approximately 1GB, encompassing benchmark datasets (SMILES strings and annotations), quantized model checkpoints, and evaluation logs, all hosted on an NVMe-backed filesystem for rapid data access.

## C.2 Evaluation Metrics

To comprehensively assess model performance, we employ the following metrics:

**Accuracy:** The proportion of correctly predicted outcomes, providing a baseline measure of overall correctness. For reaction prediction tasks (e.g., forward reaction prediction), we choose the Top-1 accuracy, which specifically means the model's highest-ranked prediction exactly matches the true product(s).

**Mean Absolute Error:** Quantifies the average magnitude of errors in continuous predictions, offering insight into precision for regression tasks (e.g., molecular property prediction).

**Scaffold Similarity:** Measured via the Tanimoto coefficient of molecular scaffolds, this evaluates structural conservation between generated and reference molecules. Values range from 0 to 1, representing scaffolds without similarity to correct scaffolds, with higher scores indicating better preservation of core frameworks.

**Improvement:** Absolute gains in target properties, reported as: Mean improvement: Average uplift across all samples. Max/min improvement: Extreme cases highlighting model potential and limitations.

**Success Rate:** The fraction of generated molecules exceeding a predefined threshold (e.g., > 0.8 for solubility), reflecting practical utility.

**Validity:** Measures the proportion of generated SMILES strings that are syntactically correct and can be successfully parsed into a chemical structure by RDKit [33].

## C.3 Count Distribution Analysis

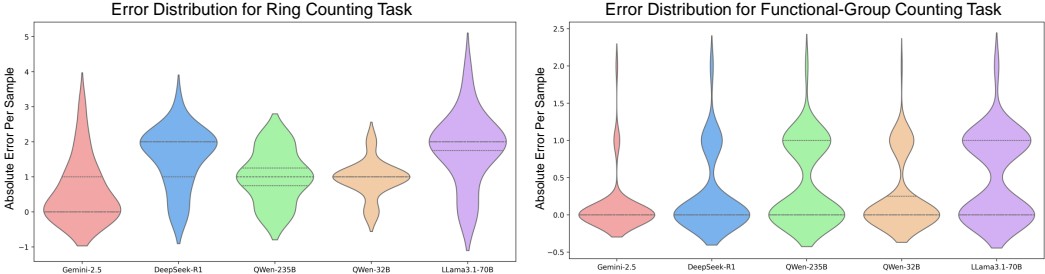

Figure 5: Error distribution analysis for ring counting and functional-group counting tasks.

For the two counting tasks under molecule understanding—ring counting and functional-group counting—we evaluated model performance using the Mean Absolute Error in the main experimental section to quantify overall accuracy. To provide a more granular analysis of LLMs' capabilities in these molecule-specific counting tasks, we further examined the error distribution across different models.

As illustrated in Fig. 5, the ring counting task proves significantly more challenging than the functional-group counting task. This is evident from the error distributions: For functional-group counting, the majority of errors fall within the 0.0–1.0 range, indicating relatively high accuracy. In contrast, ring counting exhibits higher errors, with most models (except Gemini-2.5-pro) showing an average MAE > 1.0. Gemini-2.5-pro stands out as the only model achieving consistently low errors in this task, suggesting superior structural reasoning capabilities. This disparity highlights the inherent difficulty of ring counting, which requires precise identification of cyclic structures—a more complex task

than detecting localized functional groups. The results underscore the need for further refinement of LLMs in handling intricate molecular topologies.

# D  Case Study for Tasks in ChemCoTBench

To provide a more detailed analysis of the performance of different types of LLMs across various tasks in ChemCoTBench, we supplement the quantitative findings in the Experiment section with visualizations of model outputs. In the following three subsections, we present case visualizations from distinct subtasks: molecule understanding, molecule editing, and molecule optimization.

Table 6: This is a case study for molecule understanding. We visualize the Murcko Scaffold generation task in molecule understanding because it can provide detailed information compared to number prediction tasks and correction distinguishing tasks.

| Source Molecule | GT-Scaffold | Gemini-2.5-pro | Llama3.3-70B |
|---|---|---|---|
| | 100% | 41.8% | 27.8% |
| | 100% | 38.6% | 0.0% |
| | 100% | 56.8% | 15.4% |
| | 100% | 33.3% | 13.3% |

## D.1  Case Study for Molecule Understanding

The molecule understanding task in ChemCoTBench contains three types of subtasks, including number prediction subtasks (functional-group counting and ring counting), distinguish subtasks (ring system distinguish, SMILES consistency distinguish), and scaffold generation subtask (murcko scaffold generation). To visualize the detailed molecule structure generated by different types of LLMs, we select the Murcko scaffold generation subtask as the case visualization source.

Table. 6 presents four examples featuring distinct ring structures and functional groups. Through comparative analysis, we identify two key advantages of commercial LLMs over smaller open-source LLMs:

**Superior SMILES Parsing Accuracy.** Commercial LLMs(e.g., Gemini-2.5-Pro) correctly interpret molecular SMILES structures, with predicted structures closely matching the source molecules (only 1–2 bond position errors). In contrast, open-source models like LLaMA-3.1 generate structures largely inconsistent with the source molecules.

**Robust Instruction-Following for Murcko Scaffolds.** When tasked with extracting Murcko scaffolds—defined as the maximal connected framework retaining ring systems while removing non-critical functional groups—commercial LLMs adhere to the provided instructions and generate connected scaffolds. Llama-3.1, however, often outputs fragmented substructures, highlighting its limitations in instruction comprehension.

## D.2 Case Study for Molecule Editing

The molecule editing task in ChemCoTBench contains three parts: adding a target functional group to the molecule, removing a target functional group from the molecule, and substituting a functional group with a target functional group from the molecule. In Table. 7, we visualize samples from each subtask with different types of target functional groups. Two key observations emerge from the analysis:

**Functional Group Recognition Directly Impacts Task Performance.** Gemini-2.5-Pro demonstrates high precision in functional group identification, enabling accurate molecular editing. While Qwen3-235B correctly identifies functional groups, it frequently fails to execute valid molecular modifications. LLaMA-3.1 struggles with basic functional group recognition, severely limiting its task completion capability. This trend aligns with the models' performance in the functional-group counting subtask under molecule understanding, confirming a strong correlation between recognition accuracy and downstream success.

**2D Molecular Structure Parsing Poses a Significant Challenge.** Due to the inherently linear nature of SMILES notation, LLMs generally perform well on molecules with extended one-dimensional chains. However, their accuracy declines sharply when processing complex polycyclic systems with intricate 2D topologies.

## D.3 Case Study for Molecule Optimization

Molecular Optimization Tasks involve improving three physicochemical properties (QED, Solubility, LogP) and three protein-related activation capabilities (DRD2, JNK3, GSK3-$\beta$). Since large language models perform poorly in optimizing protein-related activations, we focus on their ability to optimize physicochemical properties. Table 3 presents the optimization results of three LLMs, including Gemini-2.5-pro, Qwen3-235B, and llama3.3-70B, revealing two key observations:

**LLMs exhibit significant potential in this task.** Despite the inherent difficulty of molecular optimization, LLMs exhibit significant potential in this task. We observed that these models introduce diverse functional groups, including halogens, aldehydes, hydroxyls, and amines, indicating broad chemical adaptability. However, some modifications led to negative optimization, likely due to limited understanding of the underlying physicochemical principles—a gap that could be addressed through targeted training.

**Commercial LLMs demonstrate bolder optimization strategies compared to open-source models**. For instance, Gemini-2.5-pro frequently performs skeleton-level modifications (e.g., additions or deletions), whereas Qwen3-235B and llama3.3 tend toward conservative insertions with minimal structural changes. This contrast highlights the greater flexibility and potential of commercial LLMs in molecular optimization.

Table 7: The case study for functional-group addition, deletion, and substitution in the molecule editing task. For better comparison, we visualize the predicted results from Gemini-2.5-pro (reasoning LLM), Qwen3-235B (non-reasoning LLM), and llama3.3-70B (non-reasoning LLM) and show the outstanding chemical reasoning ability of Gemini compared to other open-sourced LLMs.

| Instruction | Source Molecule | Gemini-2.5-pro | Qwen3-235B | Llama3.3-70B |
|---|---|---|---|---|
| **Add Functional Groups** | | | | |
| Add the amide group while keeping the molecule scaffold unchanged. |  |  |  |  |
| Add the amine group while keeping the molecule scaffold unchanged. |  |  |  | **Invalid SMILES** |
| Add the benzene ring group while keeping the molecule scaffold unchanged. |  |  |  |  |
| **Delete Functional Groups** | | | | |
| Delete aldehyde group while keeping the molecule scaffold unchanged. |  |  |  |  |
| Delete hydroxyl group while keeping the molecule scaffold unchanged. |  |  |  |  |
| Delete nitro group while keeping the molecule scaffold unchanged. |  |  | **Invalid SMILES** |  |
| **Substitute Functional Groups** | | | | |
| Remove aldehyde group and add halo group for the molecule. |  |  |  | **Invalid SMILES** |
| Remove aldehyde group and add halo group for the molecule. |  |  |  |  |

Table 8: The case study for Molecule Optimizations.

| Source Molecule | Gemini-2.5-pro | QWen3-235B | Llama3.3-70B |
|---|---|---|---|
| *LogP Optimization* | | | |
|  |  $\Delta = 1.16$ |  $\Delta = 0.51$ |  $\Delta = -3.76$ |
|  |  $\Delta = 1.68$ |  $\Delta = 0.68$ |  $\Delta = -0.39$ |
|  |  $\Delta = 0.68$ |  $\Delta = 0.01$ |  $\Delta = 0.0$ |
| *QED Optimization* | | | |
|  |  $\Delta = 0.38$ |  $\Delta = 0.01$ |  $\Delta = -0.03$ |
|  |  $\Delta = 0.34$ |  $\Delta = 0.0$ |  $\Delta = -0.03$ |
| *Solubility Optimization* | | | |
|  |  $\Delta = 3.47$ |  $\Delta = 0.87$ |  $\Delta = 0.48$ |
|  |  $\Delta = 1.08$ |  $\Delta = 0.87$ |  $\Delta = 0.52$ |

## E Task Example

To better demonstrate the data structure of ChemCoTBench and the large-scale CoT dataset, we conducted visualizations of representative samples from four distinct tasks: molecule understanding, molecule editing, molecule optimization, and reaction prediction. As illustrated in Figure. 6, Figure. 7, Figure. 8, and Figure. 10, each figure presents sample cases from different tasks, with text highlighted in red indicating the chemical-specific prompt design.

---

**Question example for Molecule Understanding**

You are a chemical assistent. Please Determine whether the ring_system_scaffold is in the Molecule. Input: a molecule's SMILES string, a Ring System Scaffold. Output: yes / no.

Definition: The ring system scaffold consists of one or more cyclic (ring-shaped) molecular structures

Source Molecule: CC(C)n1cnc2c(NCc3ccc(-c4ccccc4)cc3)nc(N(CCO)CCO)nc21, IUPAC of Source Molecule: 2-[2-hydroxyethyl-[6-[(4-phenylphenyl)methylamino]-9-propan-2-ylpurin-2-yl]amino]ethanol. Ring system scaffold: c1ccc(-c2ccccc2)cc1.

Your response must be directly parsable JSON format:

{{
  "input_structure": "original input structure",

  "molecule_structure_analysis": "describe the structure of the input Molecule",

  "scaffold_analysis": "describe the ring system scaffold",

  "matching_analysis": "matching the scaffold with the molecule",

  "output": "Yes / No"
}}

DO NOT output other text except for the answer. If your response includes ```json ```, regenerate it and output ONLY the pure JSON content.

---

Figure 6: Task example for molecule understanding subtask: Ring System Counting Task.

**Question example for Molecule Editing**

You are a chemical assistant. Given the SMILES structural formula of a molecule, help me add a specified functional group and output the improved SMILES sequence of the molecule. Input: Molecule SMILES string, Functional Group Name. Output: Modified Molecule SMILES string.

Source Molecule: O=S(=O)(Cc1nc(-c2cccs2)no1)c1ccc2ccccc2n1, Instrcution: Modify the molecule by adding a aldehyde.

Your response must contain the step-by-step reasoning, and must be directly parsable JSON format:

{{
        "molecule_analysis": "[your reasoning] Analyze the functional groups and other components within the molecule",

        "function_group_introduce_strategy": "[your reasoning] Determine how and at which site the new group can be most reasonably added",

        "feasibility_analysis": "[your reasoning] Assess the chemical viability of the proposed modification",

        "output": "Modified Molecule SMILES"
}}

DO NOT output other text except for the answer. If your response includes ```json ```, regenerate it and output ONLY the pure JSON content..

Figure 7: Task example for molecule editing subtask: Functional-Group Adding Task.

**Question example for Molecule Optimization**

You are a chemical assistent,  Optimize the Source Molecule to improve the GSK3-beta property (Glycogen Synthase Kinase 3-beta Inhibition) while following a structured intermediate optimization process. IUPAC names are provided to resolve ambiguities in SMILES. For functional groups, IUPAC takes priority over SMILES. Note these key group distinctions which are difficult to distinguish (1) Piperazine (1,4-diazacyclohexane): C1CNCCN1 (2) Piperidine (azinane): C1CCNCC1 (3) Pyrrole (azole): C1=CC=CN1

Source Molecule: c1ccc(-c2cc(NCc3cccnc3)n3nccc3n2)cc1, IUPAC of Source Molecule: 5-phenyl-N-(pyridin-3-ylmethyl)pyrazolo[1,5-a]pyrimidin-7-amine.

Always output in strict, raw JSON format. Do NOT include any Markdown code block wrappers (e.g., ```json ``` or ```). Your response must be directly passable JSON format:\n
        {{
            "Structural Analysis of Source Molecule": "",
            "Property Analysis": "",
            "Limitation in Source Molecule for Property": ""
            "Optimization for Source Molecule": "",
            "Final Target Molecule": "SMILES",
        }}

DO NOT output other text except for the answer. If your response includes ```json ```, regenerate it and output ONLY the pure JSON content.

Figure 8: Task example for molecule optimization subtask: Optimizing GSK-3$\beta$ Task.

**Question example for Next Elementary-step Product Prediction**

We have one typical reaction (
   **reaction class**: 'Bromo Sonogashira coupling',
   **starting reactants**: 'CCOC(=O)C(OC(C)(C)C)c1c(C)cc2ccc(Br)cc2c1-c1ccc(Cl)cc1.C#CC(C)(C)O',
   **reagents**: 'CCN(CC)CC.C1CCOC1.CCOC(=O)C(OC(C)(C)C)c1c(C)cc2ccc(Br)cc2c1-c1ccc(Cl)cc1.C#CC(C)(C)O.[Cl-].[Cu]I.[NH4+]',
   **reaction condition**: 'Reaction with Pd coordinated with 3 or 4 ligands'
).

Here are the previous elementary reaction steps:
Elementary Step 1: {
   "reactants":
c1ccc([PH](c2ccccc2)(c2ccccc2)[Pd]([PH](c2ccccc2)(c2ccccc2)c2ccccc2)([PH](c2ccccc2)(c2ccccc2)c2ccccc2)[PH](c2ccccc2)(c2ccccc2)c2ccccc2)cc1,
   "products":
c1ccc([PH](c2ccccc2)(c2ccccc2)[Pd]([PH](c2ccccc2)(c2ccccc2)c2ccccc2)[PH](c2ccccc2)(c2ccccc2)c2ccccc2)cc1.c1ccc(P(c2ccccc2)c2ccccc2)cc1,
   "step annotation": Ligand leaving,
}

Elementary Step 2: {
   "reactants":
c1ccc([PH](c2ccccc2)(c2ccccc2)[Pd]([PH](c2ccccc2)(c2ccccc2)c2ccccc2)[PH](c2ccccc2)(c2ccccc2)c2ccccc2)cc1,
   "products":
c1ccc([PH]([Pd][PH](c2ccccc2)(c2ccccc2)c2ccccc2)(c2ccccc2)c2ccccc2)cc1.c1ccc(P(c2ccccc2)c2ccccc2)cc1,
   "step annotation": Ligand leaving,
}

Now, we want to predict the next elementary reaction step.

Currently we know the basic information:
"current_step_info": {
   "reactants": [Cu]I.C#CC(C)(C)O,
   "step annotation": Copper activation,
}

Under the same reaction condition and reagents, please give me the products of the next step element reaction. Just return the SMILES of prediction.

Your response must contains directly parsable JSON format:
{
   "pred_smi": str
}

Figure 9: Task example for mechanism prediction subtask: Next Elementary-step Product Prediction.

**Question example for Mechanism Route Selection**

For reaction class: 'Carboxylic acid + amine condensation',
under the condition of 'Condensation using BOP' and given reagents (written in SMARTS format) '[#8]=[#6]-[#8].[#7,#16,#8].[#7]-[#8]-[P+]',
which following description is the correct elementary reaction stages description, considering the mechanism of this type of reaction?

Choices:
**A**: Carboxylic acid deprotonation → Reaction of carboxylic acid and HATU/HBTU → Addition of HOBt (1-hydroxybenzotriazole) into carboxylic acid-HATU/HBTU → Amine attacks HOBt-carboxylic acid complex → Proton exchange between amide and HOBt

**B**: Proton exchange → Formation of a single bond between carboxylic acid and protonated DCC → Addition of amine (thiol) into carboxylic acid-DCC complex → Cleavage into amide and urea → Proton exchange between amide and urea

**C**: Carboxylic acid deprotonation → Reaction of carboxylic acid and CDI → Addition of imidazole into carboxylic acid-CDI → Amine attacks imidazole-carboxylic acid complex → Proton exchange between amide and imidazole

**D**: Addition of alcohol under the acidic conditions / deprotonation of alcohol → Neutralization of protonated ester / Addition of alcohol under the basic conditions

**E**: Proton exchange → Formation of a single bond between carboxylic acid and protonated DCC → Addition of HOBt (1-hydroxybenzotriazole) into carboxylic acid-DCC complex → Amine attacks HOBt-carboxylic acid complex → Proton exchange between amide and HOBt

**F**: Deprotonation of carboxylic acid → Nucleophilic substitution

**G**: Carboxylic acid deprotonation → Reaction of carboxylic acid and BOP → Addition of HOBt (1-hydroxybenzotriazole) into carboxylic acid-HATU/HBTU → Amine attacks HOBt-carboxylic acid complex → Proton exchange between amide and HOBt

**H**: Addition of amine into carboxylic acid → Deprotonation of amine → Hydroxide ion leaves

**I**: Addition to thionyl chloride → Addition of chloride → Pseudo-pericyclic expulsion of SO2, HCl → Nucleophilic addition → Nucleophilic addition → Deprotonation

**J**: Protonation of carbonyl or deprotonation of alcohol → Alcohol addition to carbonyl → Protonation or deprotonation of complex → Water or hydroxide ion leaving → Proton exchange.

Return the choice (capital letter) in JSON format:
{
    "choice": str # (e.g. 'A'/'B')
}

Figure 10: Task example for mechanism prediction subtask: Mechanism Route Selection (MechSel).

