# OpenReview forum: "Beyond Chemical QA: Evaluating LLM's Chemical Reasoning with Modular Chemical Operations"
_NeurIPS.cc/2025/Datasets_and_Benchmarks_Track — NeurIPS 2025 Datasets and Benchmarks Track poster_

### Official Review · Reviewer_LpgK · 2025-07-01

**Rating:** 4
**Confidence:** 4

**Summary:**

The authors introduce ChemCoTBench, a benchmark that shifts chemistry evaluation from factual QA to step-wise structural reasoning that includes (1) 1495 expert-audited test problems spanning four families: molecule understanding, molecule editing, property optimisation, and reaction prediction (22 subtasks).  (2) A companion 14 k chain-of-thought (CoT) corpus is distilled (via DeepSeek-R1) to facilitate training. Baselines on ~20 models highlight that commercial “thinking” LLMs outperform open-source ones, and that generic CoT distillation (Qwen-2.5) fails in chemistry unless domain-specific CoT prompts are provided.

**Dataset Code Accessibility:**

Partly

**Dataset Code Comments:**

need approval to access Huggingface datasets.

**Ethical Considerations:**

No, there are no or only very minor ethics concerns

**Final Justification:**

The author released details of how distillation are conducted and the fine-tuning experiment demonstrated that the proposed datasets do improve llm's capability in chemistry domain. Therefore, I increased my score from 3 to 4.

**Limitations Weaknesses:**

1. The paper leans on “DeepSeek-R1-distilled Qwen-2.5” to argue that generic CoT distillation fails in chemistry, yet the distillation setting is opaque, which may hinder reproducibility.

2. In Figure 4’s “CoT template” vs “CoT process” gains are a key claim, but the exact templates, token budgets, and insertion policy are missing (Appendix B.2 is only a placeholder). In addition, the train/test split setting is missing in Figure 4.

3. The dataset value could be largely boosted with a fine-tune baseline, as the author asserts the 14K CoT set will "boost open-source models" , yet they report only in-context-prompting numbers.  A simple fine-tune / LoRA baseline (e.g., Qwen-7 B trained for one epoch on the CoT set) is needed to demonstrate the dataset’s practical value.

4. Minor clarity and layout issues: Table 1 appears before Section 4; several implementation hyperparameters (e.g., which LogP/QED calculators) are unstated; and the manuscript contains typos (“substitude,” etc.).

**Strengths Contributions:**

1. The Author correctly argues that existing chem benchmarks test recall, not reasoning. The modular-operation formalism is sensible.

2. A comprehensive reasoning benchmark for chemistry tasks.

---

> ### Author Rebuttal · Authors · 2025-07-30
>
> Dear Reviewer LpgK,
>
> Many thanks to your valuable comments and questions, which help us a lot to improve our work. We address your questions as follows.
>
> ---
>
> > **Q1:** R1-Distill's closed-source distillation strategy may hinder reproducibility
>
> [A1] We appreciate your valuable suggestion. Although the DeepSeek-R1-Distilled Strategy is not open-sourced, the research community has extensively reproduced distillation-based approaches [1,2,3,4], yielding models that outperform  DeepSeek-R1-Distilled in general domains. For our evaluation, we select opensourced **S1.1-32B** [1] and **YiXin-Distill-Qwen-72B** [2] as baselines (both surpassing DeepSeek-R1-Distill-Qwen on general-domain benchmarks). In **Table.1-4**, our experiments reveal:
>
> 1. **Failure of CoT Distillation in Chemistry**: While CoT distillation (as proposed in R1-Distill-Qwen) is effective in traditional domains like math and coding, it degrades chemical domain knowledge in LLMs, impairing their ability to solve chemistry-related problems. As shown in our performance table, the vanilla Qwen-72B (without distillation) outperforms R1-Distill-72B, YiXin-Distill-72B, and S1.1-32B in chemical tasks.
>
> 2. **Degraded Cross-Domain Generalization in Reproduced Distill Models**: Although YiXin-Distill-72B and S1.1-32B claim superior performance in math and coding, they exhibit weaker generalization in chemistry compared to R1-Distill-Qwen-72B, particularly in cross-domain reasoning tasks.
>
> |   |  Func-Group-Count↓ |  Ring-Count↓ |  Murcko-Scaffold↑ |   Ring-System↑ |
> |--------|------------|---------|-----------|------|
> |  R1-Distill-72B    |  0.26  |  0.83  |  0.141  |  0.73 |
> |  YiXin-Distill-72B |  0.46  | 1.47  |  0.166  |  0.72 |
> |  S1.1-Distill-32B |  0.59  |  1.33 | 0.172 | 0.78  |
> | Qwen-72B(compare)   | **0.26** |  **0.60** |  **0.236** | **0.925** |
> **Table.1:**  Distill model performance on molecule understanding tasks.
>
> | |     Add↑  |  Delete↑   |   Substitue↑ |
> |-|------------|-----|--------|
> |R1-Distill-72B | 40 | 65   |   38.3 |
> |YiXin-Distill-72B |  50  |  55 |  49.2 |
> | S1.1-Distill-32B |  50 | 50 | 40.4 |
> | Qwen-72B |  **70**  |   **80** |  **56.7** |
> **Table.2:**  Distill model performance on molecule editing tasks.
>
> |  |  LogP↑ |  Solubility↑  |  QED↑ |  DRD2↑  |  JNK3↑ |  GSK3↑  |
> |-|----------|---------------|--------|-------------|--------|-------------|
> |R1-Distill-72B    |  0.0 |  **1.65** |  -0.06   |    0.01 |   -0.06   |  -0.03 |
> |YiXin-Distill-72B | -0.12 | 0.16  | -0.03 |  0.01 |   -0.03   |  0.0 |
> | S1.1-Distill-32B |  **0.1** | 0.21  | -0.07 |  0.0 |   -0.08   |  0.02 |
> | Qwen-72B |  -0.12 |  0.28  |  **0.03**  | **0.04** |  **0.02** | -0.01 |
> **Table.3:**  Distill model performance on molecule optimization tasks.
>
> |  |  Forward-Major↑ | Forward-By↑ | Retro↑ |   RCR↑  |  NEPP↑  |   Mech-Selection↑ |
> |-|----------|---------------|--------|-------------|--------|-------------|
> | R1-Distill-72B    | 0.67  |   0.27  |  0.39 |   0.04  |   0.68  | 0.42 |
> | YiXin-Distill-72B | **0.71** | 0.25 | 0.43 | 0.08  |   0.74  |  0.53 |
> | S1.1-Distill-32B |  0.52 | 0.19 | 0.31 | 0.06  |   0.53  |  **0.57** |
>  Qwen-72B | 0.67 | **0.28** | **0.52** | **0.18** | **0.72**  | 0.46 |
> **Table.4:**  Distill model performance on reaction prediction tasks.
>
> [1] s1: Simple test-time scaling, 2025
>
> [2] YiXin-Distill-Qwen-72B: A High-Performance Distilled Model for Mathematical and General Reasoning，2025
>
> [3] 1.4 Million Open-Source Distilled Reasoning Dataset to Empower Large Language Model Training, 2025
>
> [4] TinyR1-32B-Preview: Boosting Accuracy with Branch-Merge Distillation, 2025
>
> ---
>
> > **Q2:** Adding Implementation Details for “CoT template” vs “CoT process” Experiment inf Figure.4
>
> [A2] We sincerely thank the reviewer for the implementation details request in Figure 4 (“CoT template” vs “CoT process” Experiment). These methodological details will be explicitly added to Section 3.2 (Implementation) and Appendix B.
>
> - Token budgets: Our token budget settings follow SuperGPQA [1]. For non-thinking LLMs,  the token budget is the model's default (4K); for thinking-LLMs, it is 32K. We measured the CoT sequence lengths across all subtasks and found that most samples are shorter than 3.7K (<4K), confirming the reasonableness of our token budget strategy.
>
> - Sampling & Insertion policy: temperature=0.1 and top-p=0.5. We use the LLM's decoding strategy for text generation while constraining the output format by incorporating JSON-structured prompts.
>
> - Train/test split Details: In Figure 4, "LLM-raw," "CoT template," and "CoT process" do not require training, and their test split comprises the full ChemCoTBench dataset.  For "LLM-SFT" (details in Response-Q4.3), supervised fine-tuning (SFT)  is performed on the ChemCoT dataset with a 9:1 train-test ratio, while final evaluation still uses the entire ChemCoTBench dataset.
>
> - Exact templates: We will add all the prompt templates of “CoT template” and “CoT process” in our appendix. Here we visualize a subtask in molecule-understanding as a template example：
>
> `CoT-Template Case`
> ```
> [Think short!] You are a chemical assistant. Giving you an Input Molecule and a Fragment name and SMILES, help me count the number of fragments in the Molecule.
>             Your response must be directly parsable JSON format: \n
>             {{
>                 "fragment_structure": "fragment structure analysis",
>                 "matching_analysis": "describe and match the input Molecule with the fragment",
>                 "count": "Your Answer Number"
>             }}
> DO NOT output other text except for the answer. If your response includes ```json, regenerate it and output ONLY the pure JSON content. [Think short!]
> ```
> `CoT-Process-Case`
> ```
> [Think short!] You are a chemical assistant. Giving you an Input Molecule and a Fragment name and SMILES, help me count the number of fragments in the Molecule. To help you, I will provide you with the GROUND-TRUTH reasoning steps. FOLLOW the LEAD!!
>             Your response must be directly parsable JSON format: \n
>             {{
>                 "fragment_structure": "fragment structure analysis",
>                 "matching_analysis": "describe and match the input Molecule with the fragment",
>                 "count": "Your Answer Number"
>             }}
> DO NOT output other text except for the answer. If your response includes ```json, regenerate it and output ONLY the pure JSON content.
> ```
> [1] SuperGPQA: Scaling LLM Evaluation across 285 Graduate Disciplines, ByteDance Seed, 2025
>
> ---
>
> > **Q3:** Additional Experiments for Full-parameter SFT on ChemCoTDataset
>
> [A3] We appreciate the reviewer's suggestion and have conducted comprehensive full-parameter fine-tuning experiments on Qwen-2.5 models (1.5B, 7B, 14B, and 32B parameters).
>
> All experiments were performed on an 8×A100 cluster using the SFT framework from verl==0.4, with FSDP strategy and bfloat16 precision. The learning rate was consistently set to 2e-5 across all SFT experiments. To mitigate potential knowledge forgetting from excessive fine-tuning, we report results after exactly one epoch of training on ChemCoTDataset.
> Model Variants Explained:
>
> - Qwen-Raw: Baseline LLM performance through direct inference on ChemCoTBench
>
> - Qwen-CoT-Template: LLM inference with reasoning templates
>
> - Qwen-CoT-SFT: LLM fine-tuned on ChemCoTDataset. We fine-tune the whole dataset for 1 epoch.
>
> As shown in **Table.5**, we observe thath **ChemCoTDataset consistently enhances chemical reasoning across all model sizes**. Specifically,  CoT-SFT consistently outperforms both raw and CoT-template baselines in 6 of the 7 tasks from across all model sizes.
>
> |   |  Func-Group-Count↓ |  Ring-Count↓ |  Murcko-Scaffold↑ | Ring-System↑ | Add↑ | Delete↑ | Sub↑ |
> |--|--|--|--|--|--|--|--|
> |1.5B-Raw | 1.32 | 1.17 | 0.07 |  0.15 | 0 | 0.15 | 0.05 |
> |1.5B-CoT-Template  | 0.40 | 1.04  | 0.07 | 0.58 | 0.05 | 0.15 | 0.02 |
> |1.5B-CoT-SFT | **0.35** | **0.69**  |  **0.12**  | **0.78** | **0.20** | **0.25** | **0.07** |
> |====|====|====|====|===|====|====|====|
> |7B-Raw  | 0.43 | 1.04 | 0.09  | 0.82 | 0.15 | 0.3 | 0.15 |
> |7B-CoT-Template | 0.25 | 1.21 | 0.09 | 0.57 |  0.15 | 0.45 | 0.15 |
> |7B-CoT-SFT | 0.33 | **0.69** | **0.31**  | **0.45** | **0.40** | **0.45** | 0.15 |
> |====|====|====|====|===|====|====|====|
> |14B-Raw  | 0.42  | 1.1 | 0.11 |  0.67 | 0.35 | 0.65 | 0.2 |
> |14B-CoT-Template | 0.35 | 0.91 | 0.12 |  0.62 | 0.3 | 0.4 | 0.2 |
> |14B-CoT-SFT  | 0.41 | **0.70**  | **0.25**  | **0.63** | **0.35** | **0.70** | **0.38** |
> |====|====|====|====|===|====|====|====|
> | 32B-Raw | 0.35 | 0.95 | 0.15 | 0.60 | 0.45 | 0.55 | 0.5 |
> | 32B-CoT-Template | 0.33 | 0.74 | 0.12 | 0.70 | 0.4 | 0.65 | 0.4 |
> | 32B-CoT-SFT | **0.29**  | **0.72** | **0.17** | **0.72** | **0.55** | **0.66** | **0.53** |
> **Table.5:**  Performance of various LLMs that SFT on ChemCoTDataset.
>
> ---
>
> > **Q4:** Minor Clarity and Layout Issues
>
> [A4] We appreciate the reviewer’s careful reading and will address these points in the revision:
>
> 1. Table Placement: Table 1 will be moved to Section 4 for better flow.
>
> 2. Hyperparameters:  LogP/QED calculators are applied from the PyTDC package. We will specify the details in our appendix.
>
> 3. Typos: All spelling/grammar errors (e.g., “substitude” → “substitute”) will be corrected.
>
> ---
>
> > **Q5:** Dataset Access
>
> [A5] Thank you for your reminder about access to our ChemCoTBench and ChemCoTDataset. We have removed all restrictions from **July 15, 2025.** We are continuously updating and expanding our ChemCoTDataset, with recent additions including:
>
> - 2806 CoT samples for molecular optimization (LogP and solubility: 1,394 + 1,412 samples).
>
> - 2858 CoT samples for reaction condition recommendation (972 catalyst recommendation + 894 reagent recommendation + 992 solvent recommendation)
>
> We hope these efforts will further support the Chemical+LLM community through ChemCoTDataset and ChemCoTBench.

---

> > ### Author Response · Authors · 2025-08-05
> > **Follow-up on our Rebuttal**
> >
> > Thank you again for your time and insightful feedback. We have submitted our rebuttal and look forward to further discussion.
> >
> > We have clarified the following issues in response to your questions:
> >
> > - We provided additional experiments on open-source distalled LLMs to **comprehensively discuss the impact of the distillation strategy** on chemical tasks.
> > - We have added implementation details for “CoT template” vs “CoT process” experiment, **including token budgets, sampling policy, train-test split details, and template visualization**.
> > - We provided additional experiments **for full-parameter SFT on ChemCoTDataset**.
> > - We modified the huggingface access of ChemCoTBench and ChemCoTDataset, **all data can be downloaded without any restrictions**.
> >
> > ---
> >
> > We are confident that these clarifications and additional experimental analysis substantially strengthen our ChemCoTBench. **We would be very grateful for the opportunity to discuss any remaining questions or concerns you might have.**

---

> > ### Comment · Reviewer_LpgK · 2025-08-05
> >
> > Thank you for the detailed and thoughtful rebuttal. I appreciate the authors’ additional experiments and clarifications, particularly the efforts to benchmark multiple open-source distilled models and the new SFT results.
> >
> > However, I will maintain my original score for the following reasons:
> > 	1.	Reproducibility of Distillation Setup: While the paper makes a compelling empirical case that generic CoT distillation underperforms in chemistry, this claim rests heavily on the DeepSeek-R1-distilled Qwen model, whose distillation process remains closed-source. The rebuttal leverages alternative open-source models for support, but the lack of transparency in the original distillation setup continues to hinder reproducibility and attribution. The paper would be significantly strengthened by including a fully detailed and reproducible distillation pipeline, even for a toy model.
> > 	2.	Evaluation Leakage in SFT Results: The newly added full-parameter SFT experiments demonstrate promising gains, but the evaluation is conducted on the entire ChemCoTBench dataset, including samples seen during training. This limits the reliability of the results and overstates generalization performance. A proper held-out evaluation on a test-only subset would be essential to validate the impact of the ChemCoTDataset.

---

> > > ### Author Response · Authors · 2025-08-05
> > > **Response to Reviewer's further questions**
> > >
> > > We appreciate the reviewers' feedback. Our detailed responses follow the exact point structure of the original questions:
> > >
> > > ---
> > >
> > > > Q1: Reproducibility of R1-distill-qwen Model
> > >
> > > We clarify that our distillation experiments aim to demonstrate the weak generalizability of CoT distillation strategies in chemistry. We selected: The closed-source R1-distill-qwen along with two open-source implementations (YiXin-Distill-72B and S1.1-Distill-32B), which are community-recognized reproductions of R1-distill. These together validate our conclusions. Specifically:
> > >
> > > - R1-distill-qwen is a commercial model **whose internal process cannot be fully reproduced**. However, we have provided alternative reproductions (YiXin-Distill and S1.1-distill) that are **fully open-source and transparent**[1,2], confirming our findings.
> > >
> > > - **Designing distillation pipelines is beyond our scope**. ChemCoTBench focuses on establishing a reliable evaluation paradigm for assessing LLMs' chemical reasoning capabilities. Among our five key findings about reasoning models, one shows "distillation strategies provide no improvement in chemistry" - **but this doesn't imply we should develop new chemical-specific distillation methods, which would exceed our current work's boundaries.**
> > >
> > > - **Designing distillation strategy is not a good idea for chemical reasoning**   Based on our comprehensive evaluation of both the closed-source R1-distill-qwen and open-source implementations (yixin-distill and  s1.1-distill), we have concluded that distillation strategies do not represent a reliable approach for addressing complex chemical tasks.  Consequently, in our subsequent work, we plan to explore the use of LLM-agents as chemical knowledge supplements to tackle chemical reasoning challenges, rather than focusing on modifying distillation strategies.
> > > ---
> > >
> > > > Q2: Evaluation Leakage Concerns
> > >
> > > We confirm **there is absolutely no evaluation leakage risk** because:
> > >
> > > 1. ChemCoTBench (1.4K) and ChemCoTDataset (22K) are **completely separate datasets with no overlap**.
> > >
> > > 2. In SFT experiments:
> > >
> > >     - **Training used only ChemCoTDataset**, while small splits from ChemCoTDataset were only used to monitor potential training collapse.
> > >
> > >     - **Final evaluation used only ChemCoTBench**.
> > > ---
> > > We would be very grateful for the opportunity to discuss any remaining questions or concerns you might further have.
> > >
> > > [1] s1: Simple test-time scaling, 2025
> > >
> > > [2] YiXin-Distill-Qwen-72B: A High-Performance Distilled Model for Mathematical and General Reasoning，2025

---

> > > > ### Author Response · Authors · 2025-08-08
> > > > **Follow-up to our Further Rebuttal**
> > > >
> > > > Thank you once again for your time and insightful feedback. We have thoroughly addressed all your points and submitted a detailed rebuttal. In response to your concerns, we provide two key analyses:
> > > >
> > > > - The experiments on **R1-distill-qwen**, **YiXin-Distill**, and **S1.1-distill** sufficiently support one of our key conclusions: distill-strategy struggles with complex problems in fundamental science. Since our focus is on establishing an LLM benchmarking framework for chemical reasoning, designing chemical-distillation pipelines **falls outside the scope of this work**.
> > > >
> > > > - We clarify the train-test-split methodology in our SFT experiments and confirm that **no evaluation leakage risk exists**.
> > > >
> > > > ---
> > > >
> > > > **All feedback from other reviewers has been successfully addressed**, and they have expressed satisfaction with our revisions and the overall contribution of this work. We sincerely hope these additional clarifications and analyses resolve your concerns. Should any questions remain, we would be delighted to discuss them further.

---

> > > > > ### Comment · Reviewer_LpgK · 2025-08-09
> > > > >
> > > > > Thanks for the detailed clarification, most of my concerns are properly addressed. I will increase my score from 3 -> 4.

---

### Official Review · Reviewer_ouRW · 2025-07-02

**Rating:** 4
**Confidence:** 3

**Summary:**

The paper identifies a critical gap in how large language models (LLMs) are evaluated for chemistry tasks. Existing benchmarks mostly test factual recall (e.g., naming compounds) but fail to assess step-by-step chemical reasoning, which is essential for real-world applications like drug design and reaction engineering.

**Dataset Code Accessibility:**

Yes

**Ethical Comments:**

No information about how much money annotators earn.

**Ethical Considerations:**

Yes, there are ethics concerns that require attention by the authors

**Ethics Flags:**

["Improper research involving human subjects", "Data privacy, copyright, and consent"]

**Final Justification:**

The authors did a new set of experiments.

**Limitations Weaknesses:**

> Line 32: Unlike mathematical problems, where solutions demand explicit, verifiable steps

Please cite datasets with such problems.

> Line 38: Existing benchmarks [30, 41, 53]

No chemical datasets are mentioned in Intro. Also, popular number of datasets such as MolInstructions, MOSES, SMolInstruct, MoleculeNet are not mentioned in Related Work, no domain-specific models such as nach0 (https://huggingface.co/insilicomedicine/nach0_base), PRESTO, ChemLLM-7B, etc. in Related Work. I suggest to see survey  https://arxiv.org/pdf/2402.01439 and https://arxiv.org/pdf/2412.19994

Moreover, the authors mentioned MoleculeNet in HF readme, yet not in the paper.

The experiments mainly compare a few commercial reasoning-LLMs, general-purpose open-source LLMs, and some distilled models. Many promising domain-specific or hybrid models, such as chemistry-specialized agents beyond the tested baselines, were not included. This may limit the generalizability of the conclusions.

> Line 151: Raw molecular structures for understanding, editing, and optimization are sourced from published datasets, including PubChem [26], ChEMBL [9], ZINC [22], and Deep-Mol-Opt [16].

How many from each source?

* The experiments show that adding chemical CoT data helps, but do not systematically test how robust these gains are across LLM sizes, architectures, or different prompt engineering strategies.

> Line 182: We found that including IUPAC names helps LLMs better understand complex molecular structures, as these names offer precise details about functional groups.

What do you think it's uncommon? General-domain LLMs do not train on IUPAC-SMILES pairs in contrast with domain-specific models such as PRESTO, BioT5+, etc.

* The study shows that open-source models without domain-specific data fail to benefit much from general reasoning modes, but it does not deeply explore alternative training regimes, fine-tuning methods, or whether larger or more diverse open models could close the gap.

* The chemical accuracy assessments rely on human experts, but the experiments do not provide detailed inter-rater agreement statistics or error analyses.

**Strengths Contributions:**

* Large, domain-specific CoT dataset, carefully distilled and reviewed by multiple PhD chemists, sets a benchmark for training and evaluating LLMs. Detailed baseline results across multiple models (commercial, open-source, biomedical).
* The paper shows that adding high-quality chemical CoT data can measurably boost open-source LLMs’ performance on specialized tasks.
* The study goes beyond simple accuracy by employing domain-relevant measures like mean absolute error for counting tasks, Tanimoto similarity for scaffold extraction, and fingerprint similarity for reaction predictions.

---

> ### Author Rebuttal · Authors · 2025-07-30
>
> Dear Reviewer ouRW,
>
> Many thanks to your valuable comments and questions, which help us a lot to improve our work. We address your questions as follows.
>
> ---
> > **Q1:** Adding citation for related datasets, methods, and surveys
>
> Thank you for pointing out these omissions. We have thoroughly revised our manuscript to include a comprehensive discussion of the datasets (e.g., Mol-Instructions, MOSES) and domain-specific models (e.g., Nach0, PRESTO, ChemLLM-7B) you mentioned. This has significantly strengthened our related work section and better contextualizes ChemCoTBench's unique focus on multi-step, verifiable chemical reasoning, which distinguishes it from prior work. We have also cited the suggested survey paper ("From Words to Molecules" and "From Generalist to Specialist").
>
> ---
> > **Q2:** Baseline Expansion to chemistry-specialized LLMs
>
> We sincerely thank the reviewer for the insightful suggestion to expand our baseline comparisons. In response, we have conducted an extensive evaluation against state-of-the-art hybrid LLMs and chemistry-specialized models/agents. This not only strengthens our evaluation but also highlights the unique challenges posed by our benchmark. The newly added baselines include:
> - **Hybrid LLMs**:  We selected BioMedGPT[1] and BioMistral[2], models recognized for their strong performance on a blend of scientific and natural language tasks.
> - **Chemistry-Specialized LLMs**: We included Ether0[3], which features a specialized post-training pipeline designed for various downstream chemical tasks.
> - **Chemistry-Specialized Agents**: We evaluated leading agent systems, including ChemCrow[4] and ChemToolAgents (CTA)[5]. To ensure a fair comparison, the backbone LLM for these agents was set to open-source models.
>   - During our evaluation of the agent systems, we encountered practical limitations. For instance, ChemCrow exhibited significant challenges with instruction-following fidelity, and its strict safety controls led to a high rate of task failures on our benchmark. To maintain a meaningful comparison, its results were omitted.
>
> Our comprehensive evaluation across key tasks (molecular understanding, editing, and optimization) yielded a crucial insight: these domain-specialized models consistently underperformed the general-purpose foundation models (e.g., Qwen-72B, Mistral-24B) on the complex, multi-step reasoning tasks within ChemCoTBench. We attribute this performance gap to several factors:
> 1. **Task-Specific Overfitting**: The models' intensive fine-tuning on a few specific chemical tasks limited their generalizability across our diverse benchmark.
> 2. **Degradation of Foundational Reasoning**: Specialized post-training appears to harm the backbone LLM's core reasoning abilities, particularly on tasks outside its training distribution, such as the nuanced structural analysis required by our benchmark.
> 3. **Brittleness of Agent Systems**: Current open-source agents showed significant brittleness. Common failure points included limited toolsets unable to adapt to our tasks, low-fidelity tool-calling, and cascading errors initiated by the LLM backbone's hallucinations.
>
> ||Func-Group-Count↓|Ring-Count↓|Murcko-Scaffold↑|Ring-System↑|Add↑|Delete↑|Sub↑|
> |--|--|--|--|--|--|--|--|
> |BioMedGPT|1.6|2.43|0.18|0.53|10|12|10|
> |BioMistral|1.0|1.85|0.04|0.32|0|10|0|
> |Ether0|Failed|0.35|Failed|Failed|**94**|**76**|**78**|
> |Qwen2.5-32B-CTA|0.45|1.50|0.35|0.57|36.9|75|45.8|
> |==|==|==|==|==|==|==|==|
> |Qwen2.5-32B|0.36|0.65|0.12|0.55|50|50|48.3|
> |Mistral-24B|0.32|0.85|0.18|0.60|70|75|42|
>
> ||LogP↑|Solubility↑|QED↑|DRD2↑|JNK3↑|GSK3↑|
> |--|--|--|--|--|--|--|
> |BioMedGPT|-0.36|0.25|-0.29|0.09|-0.11|-0.08|
> |BioMistral|0.01|0.24|0.0|0.0|-0.01|-0.01|
> |Ether0|0.37|0.0|-0.37|-0.14|0.0|0.0|
> |Qwen2.5-32B-CTA|-0.26|0.18|-0.04|0.02|-0.08|0.05|
> |==|==|==|==|==|==|==|
> |Qwen2.5-32B|0.03|0.42|-0.01|0.04|-0.04|-0.02 |
> |Mistral-24B|0.05|0.30|0.03|0.01|-0.01|0.0|
>
> [1] BioMistral: A Collection of Open-Source Pretrained Large Language Models for Medical Domains, ACL 2024.
>
> [2] BiomedGPT: A Generalist Vision-Language Foundation Model for Diverse Biomedical Tasks, Nature Machine Intelligence, 2024.
>
> [3] ether0: a scientific reasoning model, dataset, and reward functions for chemistry, 2025
>
> [4] ChemCrow: Augmenting large-language models with chemistry tools, 2023
>
> [5] ChemToolAgent: The Impact of Tools on Language Agents for Chemistry Problem Solving, 2025
>
> ---
> > **Q3:** The Sample Source Analysis in ChemCoTBench and ChemCoTDataset
>
> Thanks for your valuable suggestions about the sample source analysis. The table below details the distribution of samples from different data sources in ChemCoTBench and ChemCoTDataset.
>
> `ChemCoTBench`
> |Total|PubChem|ChEMBL|ZINC|Deep-Mol-Opt|USPTO|Pistachio|
> |--|--|--|--|--|--|--|
> |1495|320 (21.4%)|100 (6.7%)|300(20%)|300(20%)|90 (6%)|385 (25.8%)|
>
> `ChemCoTDataset`
> |Total|PubChem|ChEMBL|ZINC|Deep-Mol-Opt|USPTO|Pistachio|
> |--|--|--|--|--|--|--|
> |22591|6.4k (28.3%)|4.5k (19.9%)|3.2k(14.1%)|2.4k(10.6%)|2k(9%)|4k(17.9%) |
>
> It is worth noting that our ChemCoTDataset employs an updated version, expanding the CoT data related to molecular optimization and reaction prediction. The dataset has been extended from 14,000 to 22,591 cot-samples,  and we will continue to enrich ChemCoTDataset for the benefit of the community.
>
> ---
> > **Q4 & Q6:** Adding deeper exploration on the robustness of LLM sizes, structure, prompt strategy, and fine-tuning strategy
>
> We sincerely appreciate the reviewer’s valuable suggestions. To systematically evaluate the robustness of LLMs on ChemCoTBench, we conducted comprehensive experiments across: (1) LLM sizes (1.5B, 7B, 14B, 32B) (2) Fine-tuning strategies (direct-inference, CoT-template, chemical-SFT). Key observations on molecule understanding and molecule editing tasks:
>
> 1. Scaling effect:  While larger LLM sizes generally improve performance, significant gains only occur from 1.5B→7B; diminishing returns are observed for  7B→14B→32B.
>
> 2. Strategy comparison: (1) Without chemical SFT, CoT-template shows no consistent advantage over direct inference. (2) With ChemCoTDataset-based SFT, stable improvements emerge across all LLM sizes, validating its effectiveness.
>
> ||Func-Group-Count↓|Ring-Count↓|Murcko-Scaffold↑|Ring-System↑|Add↑|Delete↑|Sub↑|
> |--|--|--|--|--|--|--|--|
> |1.5B-Direct |1.32|1.17|0.07| 0.15|0|0.15|0.05|
> |1.5B-CoT-Template|0.40|1.04|0.07|0.58|0.05|0.15|0.02|
> |1.5B-CoT-SFT|0.35|0.69|0.12|0.78|0.20|0.25|0.07|
> |==|==|==|==|==|==|==|==|
> |7B-Direct|0.43|1.04|0.09|0.82|0.15|0.3|0.15|
> |7B-CoT-Template|0.25|1.21|0.09|0.57|0.15|0.45|0.15|
> |7B-CoT-SFT|0.33|0.69|0.31|0.45|0.40|0.45|0.15|
> |==|==|==|==|==|==|==|==|
> |14B-Direct |0.42|1.1|0.11|0.67|0.35|0.65|0.2|
> |14B-CoT-Template|0.35|0.91|0.12|0.62|0.3|0.4|0.2|
> |14B-CoT-SFT|0.41|0.70|0.25|0.63|0.35|0.70|0.38|
> |==|==|==|==|==|==|==|==|
> |32B-Direct|0.35|0.95|0.15|0.60|0.45|0.55|0.5|
> |32B-CoT-Template|0.33|0.74|0.12|0.70|0.4|0.65|0.4|
> |32B-CoT-SFT|0.29|0.72|0.17|0.72|0.55|0.66|0.53|
>
> ---
> > **Q5:** Can LLMs understand IUPAC names effectively?
>
> We appreciate the reviewer's concern. IUPAC has been proven to be understandable by LLMs and can enhance chemical task performance in current works [1,2].
>
> - **Why LLM can understand IUPAC names without training**: General-domain LLMs can leverage IUPAC's systematic nomenclature as contextual clues (despite no explicit SMILES-pair training), since IUPAC rules encode structural information linguistically. Here we provide an IUPAC example: "2-chloro-4-fluorobenzoic acid", where the functional groups can be understood by general LLMs.
>
> - **Extra benefits from IUPAC-SMILES training**: Indeed, after IUPAC-SMILES training, PRESTO [3] and BioT5+ [4] have shown performance in IUPAC-to-SMILES and SMILES-to-IUPAC translation. However, in the application tasks of ChemCoTBench, which involve IUPAC-assisted molecular structure property understanding but do not involve SMILES-IUPAC translation, training on IUPAC-SMILES pairs is therefore unnecessary.
>
> [1] Assessment of fine-tuned large language models for real-world chemistry and material science applications, Chemical Science, 2025, 16, 670-684
>
> [2] Is GPT-3 all you need for machine learning for chemistry?, NeurIPS 2022, AI4Mat workshop.
>
> [3] PRESTO: Progressive Pretraining Enhances Synthetic Chemistry Outcomes, 2024
>
> [4] BioT5+: Towards Generalized Biological Understanding with IUPAC Integration and Multi-task Tuning, ACL 2024
>
> ---
> > **Q7:** The addition of inter-rater agreement statistics for human experts
>
> We fully acknowledge the reviewer's emphasis on the importance of inter-rater reliability in human evaluations. To address this, we report **Fleiss' Kappa**[1] as our inter-rater consistency metric for the 13 chemistry experts involved in constructing ChemCoTBench. Due to practical constraints (prohibitive costs of re-evaluating all tasks by all experts), we present Kappa statistics specifically for the molecular optimization subtasks, as these were the only evaluations where individual expert ratings were systematically preserved. As shown in the table below, the good Fleiss' Kappa scores (**Kappa-score > 0.6**[2]) demonstrate substantial agreement among experts across these tasks.
>
> ||LogP|Solubility|QED|DRD2|JNK3|GSK3|
> |--|--|--|--|--|--|--|
> |Kappa-score|0.88|0.71|0.69|0.62|0.69|0.6|
>
> [1] Joseph L Fleiss, Bruce Levin, Myunghee Cho Paik, et al. 1981.The measurement of interrater agreement.Statistical methods for rates and proportions, 2(212-236):22–23.
>
> [2] Reinforcement Learning from Multi-role Debates as Feedback for Bias Mitigation in LLMs
>
> ---
> > **Q8:** How much money do annotators earn?
>
> Among the 13 chemistry PhD experts participating in this study, eight of them are either employees of the collaborating company or co-authors of this work (thus requiring no additional compensation). The remaining five experts are remunerated at the standard intern rate of our collaborating company (600 RMB/person/day, ~83.6 USD).

---

> > ### Author Response · Authors · 2025-08-05
> > **Follow-up on our Rebuttal**
> >
> > Thank you again for your time and insightful feedback. We have submitted our rebuttal and look forward to further discussion.
> >
> > We have clarified the following issues in response to your questions:
> >
> > - We have **modified our citation** for related datasets, methods, and surveys.
> > - We provided **additional experiments on chemistry-specialized LLMs**, including hybrid LLMs, chemistry-specialized LLMs, and chemistry agents.
> > - We added the **sample source analysis** in ChemCoTBench and ChemCoTDataset.
> > - We added deeper exploration on the **robustness of LLM sizes, structure, prompt strategy, and fine-tuning strategy**.
> > - We have further discussed why LLMs can understand IUPAC names effectively.
> > - We have added the Kappa-score evaluation for **inter-rater agreement statistics for human experts**.
> > - We provided a detailed description of the money earned by annotators.
> >
> > ---
> >
> > We are confident that these clarifications and additional experimental analysis substantially strengthen our ChemCoTBench. **We would be very grateful for the opportunity to discuss any remaining questions or concerns you might have.**

---

> > > ### Comment · Reviewer_ouRW · 2025-08-07
> > >
> > > Thank you for the new results, I will increase my score.
> > >
> > > > fine-tuning strategies (direct-inference, CoT-template, chemical-SFT)
> > >
> > > please add all prompts as App.
> > >
> > > >IUPAC has been proven to be understandable by LLMs and can enhance chemical task performance in current works [1,2].
> > >
> > > >General-domain LLMs can leverage IUPAC's systematic nomenclature as contextual clues
> > >
> > > It would be interesting to evaluate whether model performance remains stable across different nomenclature formats and SMILES variations. For example, the recent AMORE framework (https://aclanthology.org/2024.findings-emnlp.760/) highlights that models often struggle to handle perturbations in SMILES. Including a discussion on this aspect (or see how AMORE metrics perform on your dataset if possible) would add valuable insight.

---

> > > > ### Author Response · Authors · 2025-08-07
> > > > **Official Comment by Authors**
> > > >
> > > > Thank you very much for your valuable feedback! We are happy that our responses have addressed **all your concerns**, and we will carefully incorporate the rebuttal content, the prompt template examples, and the experiments of AMORE metrics into the revised manuscript to ensure clarity and completeness.

---

> ### Comment · Area_Chair_Gtrb · 2025-08-05
>
> Dear reviewer,
>
> Did you get a chance to read the rebuttal from the authors? Does it address your questions and concerns?

---

### Official Review · Reviewer_fmbk · 2025-07-02

**Rating:** 5
**Confidence:** 4

**Summary:**

This paper presents ChemCoTBench, a benchmark of LLMs' chain-of-thought capability on chemistry knowledge. The authors argue that current chemistry-related benchmarks do not focus on reasoning ability or do not decouple fact retrieval with reasoning. To address such issues, ChemCoTBench builds a hierarchy of different chemistry tasks, including foundational topics such as SMILES understanding and molecular editing, to more complicated molecular optimization and reaction prediction. The authors evaluate several commercially available, state-of-the-art LLM models on these benchmarks and discuss strategies to enhance chemical reasoning in open-source LLMs.

**Dataset Code Accessibility:**

Partly

**Dataset Code Comments:**

The dataset is available on Hugging Face. Accessing the raw data is restricted by the time of writing this review, and I wish it would be made fully public upon acceptance.

**Ethical Considerations:**

No, there are no or only very minor ethics concerns

**Final Justification:**

I believe it is a technically strong paper of good value to the community, and should be considered for NeurIPS.

**Limitations Weaknesses:**

* One potential limitation in this work is the assumption that SMILES is the best way of handling molecules in LLMs. I agree, SMILES seems to be the default option in most cases. A brief discussion on this assumption might be nice in future revisions.

**Strengths Contributions:**

* This paper is well-motivated for the specific need of evaluating the reasoning power of LLMs in chemistry tasks. It carries great technical and scientific value and is likely to be of interest to NeurIPS audiences.
* The benchmarking tasks are well-designed, with a hierarchy of different levels of tasks (from fundamental to application). The selected chemistry tasks are novel and sound. The steps for constructing the dataset seem reasonable technically.
* The experimental evaluation is comprehensive, covering commercial LLMs and also biomolecular LLMs trained in the domain.
* The discussion of strategies to enhance chemical reasoning in open-source LLMs is interesting.

---

> ### Author Rebuttal · Authors · 2025-07-30
>
> Dear Reviewer fmbk,
>
> Many thanks to your valuable comments and questions, which help us a lot to improve our work. We address your questions as follows.
>
> ---
>
> > **Q1:** A brief discussion of why SMILES is the default option in LLM+Chem.
>
> [A1] We sincerely appreciate your valuable suggestion. Regarding the question of why LLMs predominantly use SMILES  for molecular representation, we have added a comprehensive discussion highlighting SMILES' advantages:
>
> - **Data Abundance**:  SMILES strings are widely present in pretraining corpora, enabling LLMs to better understand molecular structures and properties compared to other representations. Recent studies demonstrate that LLMs can solve basic chemical tasks even without domain-specific fine-tuning [1] or serve as effective molecular encoders [2].
>
> - **Self-Corrective Mechanism**:  While SMILES may occasionally generate invalid structures, work by [3]  shows that this property is not a drawback but rather a feature that improves model robustness. Their experiments demonstrate that LLMs pretrained on SMILES outperform those trained on inherently valid alternatives (e.g., SELFIES).
>
> We acknowledge emerging alternatives (t-SMILES[4], SELFIES, IUPAC names, etc.), but their adoption is limited by niche usage, excessive length,  or data scarcity. Nevertheless, auxiliary inputs, such as IUPAC names, can enhance chemical understanding. In Section 3.3 (LLM-based CoT Evaluation), we incorporate IUPAC names into ChemCoTBench/Dataset to improve functional group recognition.
>
> [1] What can large language models do in chemistry? a comprehensive benchmark on eight tasks, NeurIPS 2023
>
> [2] Can large language models understand molecules? BMC Bioinformatics (2024)
>
> [3] Invalid SMILES are beneficial rather than detrimental to chemical language models, Michael A. Skinnider, Nature Machine Intelligence, 6, pages 437–448 (2024)
>
> [4] t-SMILES: a fragment-based molecular representation framework for de novo ligand design, Nature Communications, 15, 4993 (2024)
>
> ---
>
> > **Q2:** Access of ChemCoTBench
>
> [A2] Thanks for your reminder about the access to our ChemCoTBench and ChemCoTDataset. We **have removed all restrictions** from July 15, 2025. We are continuously updating and expanding our ChemCoTDataset, with recent additions including:
> - 2806 CoT samples for molecular optimization (LogP and solubility: 1,394 + 1,412 samples).
> - 2858 CoT samples for reaction condition recommendation (972 catalyst recommendation + 894 reagent recommendation + 992 solvent recommendation)
>
> We hope these efforts will further support the Chemical+LLM community through ChemCoTDataset and ChemCoTBench.

---

> > ### Comment · Reviewer_fmbk · 2025-08-04
> >
> > Thank you for the clarification; I have no further questions.

---

> > > ### Author Response · Authors · 2025-08-07
> > > **Official Comment by Author**
> > >
> > > Thank you very much for your valuable feedback! We are glad that our responses **have addressed all your concerns**, and we will carefully incorporate the rebuttal content into the revised manuscript to ensure clarity and completeness.

---

### Official Review · Reviewer_TYNi · 2025-07-03

**Rating:** 5
**Confidence:** 4

**Summary:**

ChemCoTBench is a benchmark designed to assess Llms' ability to perform step-by-step chemical reasoning. They frame chemistry problems as sequences of modular operations on SMILES strings, addition, deletion, substitution, spanning four task categories: basic molecular understanding, simple editing, property optimization, and reaction prediction. The paper provides 1,495 expert annotated CoT (chain-of-thought) examples for evaluation and a complementary 14,000-sample dataset for training. Through systematic experiments on both closed (e.g. GPT-4, Claude) and open-source models (e.g. Llama, Qwen), the authors show that (a) advanced “thinking” models outperform their vanilla counterparts, (b) open models need domain-specific CoT data to unlock chemistry reasoning, and (c) transfers to complex tasks. It is an important benchmarking foundation for measuring and improving Llms’ real-world chemical problem-solving.

**Additional Feedback:**

--> Have you considered integrating molecular graphs or images (2D sketches) so models can reason beyond linear SMILES?
--> Memorization v.s. Reasoning: What controls or analyses could help disentangle an LLM’s recall of known reactions from its genuine ability to plan novel sequences of modular operations?
--> agent-based/tool-use: Could future iterations allow LLMs to call external chemistry toolkits (e.g. RDKit, retrosynthesis planners) as part of the CoT, and how would that affect both benchmark design and model performance?

**Dataset Code Accessibility:**

Yes

**Ethical Considerations:**

No, there are no or only very minor ethics concerns

**Final Justification:**

The rebuttal from the authors have convinced me and addressed most of my concerns. Thus, I suggest accepting the paper.

**Limitations Weaknesses:**

- The labor-intensive annotation process (~1,800 hours) may be hard to extend to broader chemistries (e.g. biopolymers, inorganic systems).
- Back to the point above, focusing on small organic molecules and common reactions leaves gaps in stereochemistry, complex ring systems, and bio- or materials chemistry.
- Exact-match or fingerprint-similarity metrics can penalize valid alternative solutions or creative syntheses, underestimating model ingenuity or , worse, biasing the development of LLMs.
- Proprietary models may have seen benchmark source data (e.g. USPTO patents) during pretraining, conflating memorization with genuine reasoning.

**Strengths Contributions:**

- By decomposing problems into a small set of modular edits, the benchmark forces models to “show reasononing” making each intermediate step checkable and interpretable.
- The progression from simple (group counting) to advanced (mechanism prediction) tasks provides diagnostic insight into where a model’s reasoning pipeline breaks down.
- Dual LLM + human review add credibility to the benchmark dataset and ensuring chemical validity.
- The 1.5k evaluation set paired with a 14k CoT training corpus both assesses and improves models. This can drive better community adoption.
- Tasks represent real medicinal chemistry and organic synthesis challenges (e.g. optimizing drug-like properties, predicting reaction by-products). It is a good addition to the mix of benchmark dataset we have now.

---

> ### Author Rebuttal · Authors · 2025-07-30
>
> Dear Reviewer TYNi,
>
> Many thanks to your valuable comments and questions, which help us a lot to improve our work. We address your questions as follows.
>
> ---
>
> >  **Q1:** The labor-intensive annotation process (~1,800 hours) may be hard to extend to broader chemistries
>
> We appreciate the reviewer’s concern regarding labor-intensive annotation. The reported 1,800-hour effort~(Line 97) encompasses both human expertise and LLM-assisted sample generation, with the following breakdown:
>
> - Human Expertise (30% effort):  **13 chemistry** PhDs each dedicated **1–2 days** to refine task formulations in the scientific domain and curate high-quality LLM-generated samples.
>
> - LLM-Assisted Generation (70% effort): The majority of time was allocated to scalable, model-driven sample synthesis.
>
> This human-in-the-loop pipeline proves to be a highly efficient and scalable paradigm for creating high-quality, specialized CoT datasets. It allows us to leverage expert knowledge for defining complex tasks while using LLMs for scalable data generation, making it readily extensible to broader chemistries as suggested.
>
> ---
>
> >  **Q2:** ChemCoTBench focuses on small organic molecules and common reactions, leaving opportunities for other domains.
>
> We agree with the reviewer that our ChemCoTBench primarily covers small organic molecules and reactions. which indeed leaves opportunities for extension to stereochemistry, biology, and material chemistry. However, our ChemCoTBench has domain expansion potential for two reasons:
>
> 1. **Foundational Priority**: Small molecules and common reactions represent the highest-frequency knowledge in chemistry, biology, and material domains. Thus, ChemCoTBench is a logical starting point for establishing evaluation benchmarks on biology and material domains.
>
> 2. **Annotation Scalability**: The modular design of our annotation pipeline~(Fig. 3) explicitly decouples domain expertise from task logic. By reusing the same workflow but swapping in biopolymer/materials experts, the system can expand to new domains without redesign, only requiring incremental annotation time that **much less than 1800h.**
>
> We appreciate the reviewer's suggestion regarding extension to other domains, and we plan to expand our work to crystal structure prediction tasks in materials science, as well as protein-ligand affinity prediction in the protein field. Here, in **TABLE.1**, we provide a detailed construction roadmap of expanding our ChemCoTBench to the crystal structure prediction.
>
>  | Crystal Dataset   |    Crystal Metrics   |  General-models | Specialist-models |
>  |--|--|--|--|
>  | The Materials Project, ICSD(crystal database) | CIF similarity, $E_{hull}$, Stability(M3GNet[1], VASP[2]) | GPT, Gemini, Claude, etc | MatBERT[3], SentMatBERT_MNR[4], Crystall-llama[5], etc |
> **TABLE.1:**  Construction Pipeline for expanding ChemCoTBench to the crystal domain.
>
> [1] A universal graph deep learning interatomic potential for the periodic table, nature computational science, 2022
>
> [2] Ab-initio simulations of materials using VASP: Density-functional theory and beyond, Computational Chemistry, 2008.
>
> [3] The Impact of Domain-Specific Pre-Training on Named Entity Recognition Tasks in Materials Science, 2021
>
> [4] From Tokens to Materials: Leveraging Language Models for Scientific Discovery, 2024
>
> [5] Fine-Tuned Language Models Generate Stable Inorganic Materials as Text, ICLR 204
>
> ---
>
> >  **Q3:** Exact-Match Metric may penalize creative syntheses.
>
> We acknowledge the reviewer’s concern regarding the importance of preserving LLM creativity in metric design. However, ChemCoTBench employs a comprehensive evaluation framework to rigorously assess LLM outputs:
>
> - **MAE and Accuracy** on molecule-understanding tasks:  The answers in molecule-understanding tasks are definitive, especially for subtasks such as determining the presence of ring systems or functional groups, which are evaluated using yes/no questions.  Therefore, the exact match metric does not affect the assessment of LLM  creativity in this part.
>
> - **Flexible-match using Chemical-tools** on molecule editing and optimization:  For these two molecular design tasks, we fully consider flexible evaluation for LLMs. In molecule editing, we use RDKit to verify whether molecular functional groups are added or deleted according to the instructions, thereby calculating the correct rate. In molecule optimization, we employ PyTDC as a property predictor to evaluate the success rate of molecular optimization. Our specific implementation is provided in the open-source code at `./baselines/eval/eval_metric.py` from our GitHub repo.
>
> - **Average FTS(RDK-FTS, MACCS-FTS, Morgan-FTS) and Top-1 metric** on reaction prediction tasks:  Evaluating the feasibility of chemical reaction equations remains challenging. Thus, our metric follows the common reaction metrics applied in various chemical LLMs, including Mol-Instructions[1], Instruct-Mol[2], and BioT5+[3]. Specifically, our metric includes top-1 accuracy and average fingerprint similarity (FTS), which is the average of RDK-FTS, maccs-FTS, and morgan-FTS. We also evaluated the model from the perspective of SMILES sequences (using BLEU and Levenshtein distance) and molecular validity.  Due to space constraints, these results are not included in the main text. Instead, in **Table.2**, we present a subset of evaluation results on the forward major production prediction task for illustration:
>
> | Baseline | Top-1↑ | BLEU↑ | Levenshtein↓ | RDK-FTS↑ | maccs-FTS↑ | morgan-FTS↑ | Validity↑ | FTS↑ |
> |--|--|--|--|--|--|--|--|--|
> |DeepSeek-R1 | 0.48 | 0.79 |5.82 |0.77 |0.83 |0.75 | 0.91|0.71 |
> | claude-3-7-sonnet           | 0.73 | 0.91 |  2.42 | 0.90 | 0.93 | 0.89 | 0.96 | 0.87 |
> | gemini-2.5-pro              | 0.72 | 0.91 |  3.05 | 0.90 | 0.94 | 0.89 | 0.98 | 0.89 |
> | o1-mini                     | 0.26 | 0.49 |  6.77 | 0.48 | 0.55 | 0.47 | 0.62 | 0.31 |
> | o3-mini                     | 0.52 | 0.81 |  4.63 | 0.78 | 0.83 | 0.77 | 0.89 | 0.71 |
> | qwen3-235b-a22b-reason      | 0.03 | 0.68 | 19.04 | 0.59 | 0.68 | 0.53 | 0.90 | 0.54 |
> | llama3.3-nemotron-49B-reason| 0.09 | 0.70 |  9.33 | 0.31 | 0.38 | 0.30 | 0.54 | 0.18 |
> | qwen3-32B-reason            | 0.11 | 0.74 | 11.90 | 0.48 | 0.53 | 0.43 | 0.68 | 0.33 |
> | ether0                      | 0.70 | 0.72 | 13.36 | 0.78 | 0.81 | 0.77 | 0.86 | 0.68 |
> | DeepSeek-v3                 | 0.36 | 0.72 |  9.47 | 0.70 | 0.77 | 0.68 | 0.87 | 0.62 |
> **TABLE.2:**  Evaluation results on the Reaction Forward Major Production Prediction Task.
>
> [1] Mol-Instructions: A Large-scale Biomolecular Instruction Dataset for LLMs, ICLR 2024
>
> [2] InstructMol: Multi-Modal Integration for Building a Versatile and Reliable Molecular Assistant in Drug Discovery, CoLING, 2025
>
> [3] Biot5+: Towards generalized biological understanding with iupac integration and multitask tuning, EMNLP 2023
>
> ---
>
> >  **Q4:** The data leakage of proprietary models during pretraining may conflate memorization with genuine reasoning.
>
> We appreciate the reviewer's concern regarding data leakage, which indeed remains a persistent challenge for LLMs in both general and scientific domains [1,2,3]. However, **ChemCoTBench demonstrates stronger robustness to this issue** due to two key design features:
>
> 1. **Complex Reasoning Design** mitigates data leakage scenarios: While data leakage may enable LLMs to memorize question-answer pairs in factual retrieval tasks, our benchmark requires correct reasoning-path execution to arrive at solutions. This multi-step reasoning paradigm significantly mitigates the impact of potential memorization, enhancing robustness against data leakage scenarios.
>
> 2. **Comprehensive Data Reshaping** mitigates data leakage scenarios:
> All external data (e.g., from USPTO and Pistachio for reaction prediction tasks) undergo rigorous transformation through: (1) Specialized chemical tools [4,5] for detailed equation annotation. (2) Contextual prompt engineering to obscure original data signatures. This processing pipeline effectively breaks potential memorization patterns while preserving scientific validity.
>
> [1] Leak, Cheat, Repeat: Data Contamination and Evaluation Malpractices in Closed-Source LLMs. 2024
>
> [2] Beware of Data Leakage from Protein LLM Pretraining, 2024
>
> [3] Reasoning or Memorization Unreliable Results of Reinforcement Learning due to Data Contamination
>
> [4] Reproducing Reaction Mechanisms with Machine-Learning Models Trained on a Large-Scale Mechanistic Dataset, 2024, Angewandte Chemie.
>
> [5] Electron flow matching for generative reaction mechanism prediction obeying conservation law, 2025.

---

> > ### Author Response · Authors · 2025-08-05
> > **Follow-up on our Rebuttal**
> >
> > Thank you again for your time and insightful feedback. We have submitted our rebuttal and look forward to further discussion.
> >
> > We have clarified the following issues in response to your questions:
> >
> > - We added the **Annotation Cost Details** of our ChemCoTBench and ChemCoTDataset
> > - We have further discussed the **domain expansion potential of our ChemCoTBench** by adding a detailed proposal for expanding our benchmark to materials science.
> > - We provided a **detailed description of the diverse metrics** of our ChemCoTBench.
> > - We have further discussed the **potential issue of data leakage in LLMs**.
> >
> > ---
> >
> > We are confident that these clarifications and additional experimental analysis substantially strengthen our ChemCoTBench. **We would be very grateful for the opportunity to discuss any remaining questions or concerns you might have.**

---

> > > ### Comment · Reviewer_TYNi · 2025-08-07
> > > **Concerns addressed**
> > >
> > > Thanks for the effort in addressing my concerns. I have read your rebuttals and been convinced. Thus, I decided to increase my previous rating.

---

> > > > ### Author Response · Authors · 2025-08-07
> > > > **Official Comment by Authors**
> > > >
> > > > Thank you very much for your valuable feedback! We are glad that our responses have addressed **all your concerns**, and we will carefully incorporate the rebuttal content into the revised manuscript to ensure clarity and completeness.

---

### Author Response · Authors · 2025-08-04
**General Response**

We sincerely thank the reviewers for their detailed and valuable comments. **All reviewers** (TYNi, fmbk, ouRW,  LpgK) appreciated our novel approach of decoupling complex chemical tasks into chemical operations. **All reviewers (TYNi, fmbk, ouRW,  LpgK) acknowledged the contributions of ChemCoTBench to the LLM community**, particularly in terms of:

- **Task comprehensiveness** (LpgK, ouRW)

- **Reliability of chemical annotations** (TYNi, fmbk, ouRW)

- Representation of real-world medicinal chemistry and organic synthesis challenges, making it **a good addition to existing benchmarks** (TYNi)

- **Significant technical and scientific value**, likely to interest the NeurIPS audience (fmbk).

Moreover, our proposed ChemCoTDataset, **the first large-scale, trainable chemical CoT dataset**, is shown to enhance LLM performance on chemical tasks (TYNi, ouRW). From an experimental perspective, our analysis was deemed **thorough and insightful (fmbk, ouRW)**.

Based on these comments, we conclude some noteworthy replies for the reviewers, including:
- [Reviewer TYNi] We added the Annotation Cost Details of our ChemCoTBench and ChemCoTDataset
- [Reviewer TYNi] We have further discussed the domain expansion potential of our ChemCoTBench, by adding a detailed proposal for expanding our benchmark to materials science.
- [Reviewer TYNi] We provided a detailed description of the diverse metric of our ChemCoTBench.
- [Reviewer TYNi] We have further discussed the potential issue of data leakage in LLMs.
- [Reviewer fmbk] We added a brief discussion of why SMILES is the default option in chemical LLMs.
- [Reviewer fmbk, LpgK] We modified the huggingface access of ChemCoTBench and ChemCoTDataset, all data can be downloaded without any restrictions.
- [Reviewer ouRW] We have modified our citation for related datasets, methods, and surveys.
- [Reviewer ouRW] We provided additional experiments on chemistry-specialized LLMs, including hybrid LLMs, chemistry-specialized LLMs, and chemistry-agent.
- [Reviewer ouRW] We added the sample source analysis in ChemCoTBench and ChemCoTDataset.
- [Reviewer ouRW] We added deeper exploration on the robustness of LLM sizes, structure, prompt strategy, and fine-tuning strategy
- [Reviewer ouRW] We have further discussed why LLMs can understand IUPAC names effectively.
- [Reviewer ouRW] We have added the Kappa-score evaluation for inter-rater agreement statistics for human experts.
- [Reviewer ouRW] We provided a detailed description of the money earned by annotators.
- [Reviewer LpgK] We provided additional experiments on open-source distalled LLMs to comprehensively discuss the impact of distillation strategy on chemical tasks.
- [Reviewer LpgK] We have added implementation details for “CoT template” vs “CoT process” experiment in Figure.4
- [Reviewer LpgK] We provided additional experiments for full-parameter SFT on ChemCoTDataset

---

### Decision · Program_Chairs · 2025-09-18

**Decision:**

Accept (poster)

**Comment:**

Chain-of-Though (CoT) reasoning has been shown to improve large language model (LLM) performance in mathematics and coding. There is a potential that CoT can also be useful for chemistry related tasks processed by LLM. Existing benchmarks and datasets only focus on factual retrieval and for evaluating complicated tasks it is often ambiguous how much performance stems from factual retrieval and how much performance stems from reasoning. To narrow this gap, ChemCoTBench was introduced to decompose the solution of chemistry problems, including molecule understanding, molecule editing, molecule optimization, and reaction prediction, into step-by-step workflows. A high-quality human expert reviewed/edited 1.5k evaluation set paired with a 14k CoT training corpus was prepared. Existing reasoning and non-reasoning LLM models have been experimented in the ChemCoTBench evaluation, demonstrating there is room for existing models to improve in the domain of complex chemical reasoning.

All the reviewers consistently think highly of the contribution of ChemCoTBench to the LLM and chemistry communities, because of its novelty, annotation credibility, and task comprehensiveness. There are some critical questions from the reviewers about evaluating the chemistry-specialized LLM, fine-tuning LLMs using the 14k CoT training corpus, and the authors have performed additional experiments and adequately addressed them. Therefore, I am recommending the paper for acceptance.

However, it should be noted that there are some drawbacks of the paper in my own opinion, which have not been discussed very much in the rebuttal period, because the reviewers were not actively involved in the conversation with me. Lots of details related to the labeling process, especially the prompts used for LLM-assisted CoT annotation and the criteria of evaluating those prompts were missing, preventing the community from using the similar approach to create datasets for other closely related domains. In addition, even though the evaluation dataset of ChemCoTBench was step-wise annotated for reasoning, the LLMs were only evaluated on the final problem outputs. Because the expert-refined reasoning was not used in the LLM evaluation, it is confusing to the readers why thousands of hours were spent on the ChemCoTBench reasoning refinement and how exactly the true LLM reasoning capability can be evaluated.